# Intermediate progenitors support migration of neural stem cells into dentate gyrus outer neurogenic niches

Branden R Nelson[1,2]*, Rebecca D Hodge[1†], Ray AM Daza[1‡],
Prem Prakash Tripathi[1§], Sebastian J Arnold[3,4], Kathleen J Millen[1,5],
Robert F Hevner[1,2‡]*

[1]Center for Integrative Brain Research, Seattle Children's Research Institute, Seattle, United States; [2]Department of Neurological Surgery, University of Washington, Seattle, United States; [3]Institute of Experimental and Clinical Pharmacology and Toxicology, Freiburg, Germany; [4]Signaling Research Centers BIOSS and CIBSS, Faculty of Medicine, University of Freiburg, Freiburg, Germany; [5]Department of Pediatrics, University of Washington, Seattle, United States

**\*For correspondence:**
branden.nelson@seattlechildrens.org (BRN);
rhevner@ucsd.edu (RFH)

**Present address:** [†]Allen Institute for Brain Science, Seattle, United States; [‡]Department of Pathology, University of California San Diego, La Jolla, United States; [§]CSIR-Indian Institute of Chemical Biology, Kolkata, India

**Competing interests:** The authors declare that no competing interests exist.

**Abstract** The hippocampal dentate gyrus (DG) is a unique brain region maintaining neural stem cells (NCSs) and neurogenesis into adulthood. We used multiphoton imaging to visualize genetically defined progenitor subpopulations in live slices across key stages of mouse DG development, testing decades old static models of DG formation with molecular identification, genetic-lineage tracing, and mutant analyses. We found novel progenitor migrations, timings, dynamic cell-cell interactions, signaling activities, and routes underlie mosaic DG formation. Intermediate progenitors (IPs, Tbr2+) pioneered migrations, supporting and guiding later emigrating NSCs (Sox9+) through multiple transient zones prior to converging at the nascent outer adult niche in a dynamic settling process, generating all prenatal and postnatal granule neurons in defined spatiotemporal order. IPs (Dll1+) extensively targeted contacts to mitotic NSCs (Notch active), revealing a substrate for cell-cell contact support during migrations, a developmental feature maintained in adults. Mouse DG formation shares conserved features of human neocortical expansion.

## Introduction

Complex tissue formation and morphogenesis often require stem cells to migrate away from their site of origin through new territory and over long distances to assemble de novo structures, such as the prototypical migrating neural crest and its diverse lineages. Indeed, revelations regarding how evolutionarily regulated progenitor cell migrations expand and shape the neocortex have transformed our view of human brain development (*Hansen et al., 2010*; *Nowakowski et al., 2016*). Moreover, certain human brain structures maintain neural stem cell (NSC) and progenitor cell populations through late fetal-to-postnatal development (e.g. cerebellum) (*Leto et al., 2016*; *Marzban et al., 2014*), and even later ages, such as the hippocampal neurogenic niche located in the dentate gyrus (DG) (*Boldrini et al., 2018*; *Moreno-Jiménez et al., 2019*; *Snyder, 2019*; *Sorrells et al., 2018*). How stem and progenitor cell migrations are guided and maintained during extensive migrations, and how their progeny assemble new structures are topics of keen interest across many developing tissues, since congenital defects and environmental insults can impact these processes.

DG formation in rodents provides a general framework for investigating how migrating neural progenitor cells are maintained and guided during their long migrations to construct new 'outer'

(abventricular) neurogenic niches and assemble critical tissue architecture (*Berg et al., 2018*). However, compared to studies of adult DG neurogenesis, studies of DG development are relatively limited. Nonetheless, progenitor cell migration has been a prominently proposed feature, marked classically by formation of the dentate migratory stream (DMS) from the dentate neuroepithelial stem zone (DNe) to the developing DG (*Figure 1A*, *Figure 1—figure supplement 1*). Progenitor cells in the DMS are furthermore distributed in multiple transient (but distinct) niches, presumably migrating to the nascent DG before finally settling in the subgranular zone (SGZ), an enduring niche that persists to youth or adulthood (*Figure 1A*; *Altman and Bayer, 1990a*; *Altman and Bayer, 1990b*; *Hodge et al., 2013*; *Li et al., 2009*; *Nicola et al., 2015*). The distinct roles and contributions of IPs and NSCs to DG morphogenesis, and the cellular interactions supporting NSC migration and maintenance in the DMS and other DG niches, are unknown. Moreover, exactly how multiple streams of migrating progenitor cells differentiate into granule neurons (GNs) and construct the granule cell layer (GCL) in an 'outside-in' mosaic fashion are not clear (*Angevine, 1965*; *Martin et al., 2002*).

Generally, NSCs require activated Notch signaling to prevent neuronal differentiation and maintain stemness (*Breunig and Nelson, 2020*; *Lui et al., 2011*; *Zhang et al., 2018*). In the mouse neocortex, IPs produce Notch ligands (e.g Dll1) that activate Notch receptors in neighboring NSCs to drive and maintain Notch signaling (*Kawaguchi et al., 2008*; *Mizutani et al., 2007*; *Nelson et al., 2013*; *Yoon et al., 2008*). Outer NSCs and IPs are especially abundant during neocortical development in higher order gyrencephalic mammalian cortices (*Betizeau et al., 2013*; *Fietz et al., 2010*; *Hansen et al., 2010*; *Lui et al., 2011*). Recent studies suggest outer NSCs are regulated by evolutionary modifications in Notch signaling that control progenitor diversity and proliferation (*Fiddes et al., 2018*; *Suzuki et al., 2018*). Local amplification of IPs and Notch-active NSCs is thought to be an important initial step preceding outer migrations that underlie gyrus formation and expansion during neocortical development (*Llinares-Benadero and Borrell, 2019*). Similarly, we hypothesized that IPs might interact with NSCs during DG formation, supporting NSCs during their lengthy outer migrations to establish and maintain a new adult neurogenic niche. We tested this hypothesis using live-cell multiphoton time-lapse microscopy and image analyses of developing mouse DG in slice cultures, in conjunction with cell-type specific IP and NSC markers and transgenic reporters, and with conditional knockout and fate-mapping lineage studies.

## Results

### Increased IP production and high notch activity distinguish the early DNe stem zone from adjacent regions

In mice, the DMS initially forms on embryonic day (E) 14.5 by migration of progenitor cells and immature neurons towards the pial surface and hippocampal fissure (*Figure 1A*, *Figure 1—figure supplement 1*; *Altman and Bayer, 1990a*; *Altman and Bayer, 1990b*; *Hodge et al., 2013*; *Li et al., 2009*), although the exact composition and timing of cell types emerging from the DG neuroepthelial (DNe) stem zone are not clear. We examined gene expression databases and earlier stages to better define when and where IPs, NSCs, and the DMS first arise (*Figure 1A*, *Figure 1—figure supplement 1*): Allen Developing Mouse Brain Atlas (2008) http://developingmouse.brain-map.org/. We used well-defined regional (*Lef1*, *Wnt2b*, *Msx1*, *Cxcl12*, *Lmx1a*) and cell-type specific markers (*Sox2*, *Eomes*, *Neurod1*, *Trp73*, *Reln*) to identify the early DNe stem zone (*Figure 1A*, *Figure 1—figure supplements 1* and *2*). Surprisingly, substantial *Eomes+* IP production marked a discrete DNe ventricular patch even at early stages (E11.5, *Figure 1—figure supplement 1B*). Moreover, *Eomes+* IPs had already accumulated in an outer DNe region near the pial surface, defining the DMS origin by E11.5 that was even more apparent by E13.5 (*Figure 1—figure supplement 1B*). Importantly, initial IP production and accumulation in the DNe were readily distinguished from flanking hippocampal CA field and fimbria/choroid plexus stem zones at E13.5, the latter of which is marked by *Lmx1a+* progenitors that give rise to *Trp73+ Reln+* Cajal-Retzius cells (CRs) (*Figure 1—figure supplement 2A*; *Anstötz and Maccaferri, 2020*; *Hodge et al., 2013*). Pax6/Tbr2/Reelin triple immunofluorescence at E14.5 reveals distinct DNe derived Tbr2+ IPs (Pax6+/- and Reelin-) in the DMS (*Figure 1—figure supplement 2B*), while transient Tbr2 expression overlap is observed as CRs differentiate (Tbr2+ Reelin+) and emerge from the more medial hem/fimbria (*Figure 1—figure supplement 2C*).

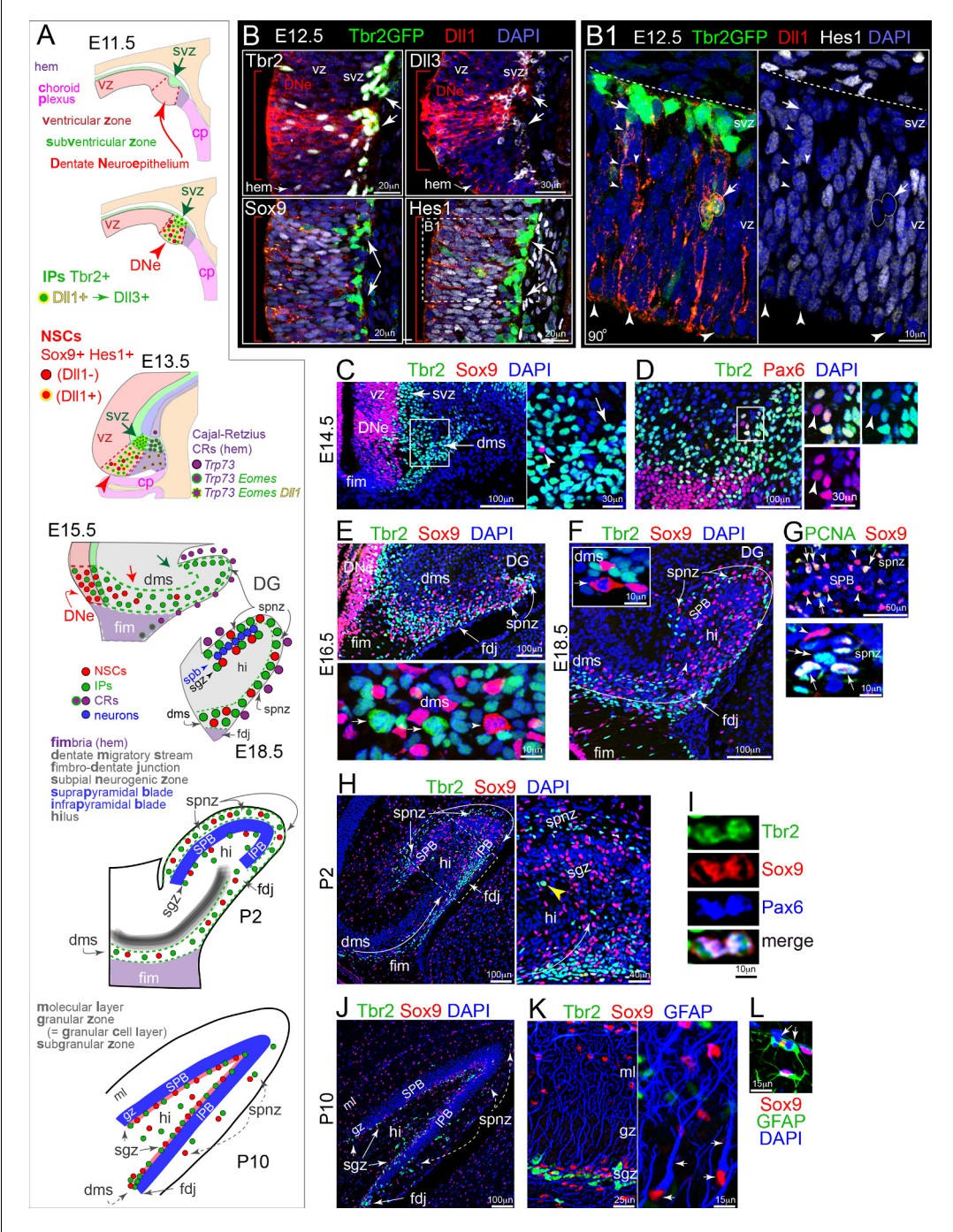

**Figure 1.** Development of DG: Notch activation and sequential IP-NSC migrations. (**A**) Schematics of DG development and cell migrations (see also *Figure 1—figure supplement 1*). (**B**) E12.5 coronal sections immunolabeled for Dll1 and IP and NSC markers: the DNe is marked by intense Dll1+ immunoreactivity (red arrowheads and brackets) with increased IPs (top: Dll1+,Tbr2+, Dll3+, Tbr2GFP+ TG) and Notch active NSCs (bottom: Sox9+ Hes1+). (**B1**) Boxed region from Hes1 rotated 90° shows Dll1+/Hes1-/GFP+ IPs dividing in basal VZ and extending long-range branched processes from SVZ into VZ (arrows), and mitotic-Hes1$_{low}$ NSCs at the VZ surface encased by neighboring Dll1 puncta (arrowheads). (**C**) On E14.5, Tbr2+ IPs pioneered the DMS (inset arrow) in advance of Sox9+ NSCs (inset arrowhead). (**D**) Most cells in E14.5 DMS were IPs (Tbr2+); NSCs (Pax6+/Tbr2-) were sparse (inset arrowhead). (**E**) By E16.5, the DMS contained Sox9+ NSCs, as well as Tbr2+ IPs: the FDJ and SPNZ had developed and likewise contained both progenitor types. Bottom panel: mitotic Tbr2+ IPs (arrows) and Sox9+ NSC (arrowhead) in the DMS. (**F**) By E18.5, transient neurogenic niches (DMS→FDJ→SPNZ,Hi) were well defined and the SPB was visible. Note Sox9+ mitosis in DMS (inset). (**G**) The E18.5 SPB (top panel) was delimited by rows of NSCs (Sox9+) in the SPNZ and nascent SGZ. Some NSCs were proliferative (PCNA+), others quiescent (PCNA-). PCNA+/Sox9- proliferative IPs (bottom panel, double arrow) were also seen in E18.5 SPNZ. (**H**) On P2, Tbr2+ IPs and Sox+ NSCs were abundant in the DMS, SPNZ, Hi, and FDJ; and

*Figure 1 continued on next page*

*Figure 1 continued*

IPB formation was apparent. Boxed area shown at higher magnification in right panel. Note mitotic Tbr2+ IP in the nascent SGZ (yellow arrowhead). (**I**) Rare triple-labeled Tbr2+/Sox9+/Pax6+ mitotic figure in the FDJ, in transition from NSC to IP. (**J**) By P10, the SGZ contained abundant Tbr2+ IPs and Sox9+ NSCs (arrows). Transient niches (SPNZ, FDJ, Hi) were declining. Sox9+ astrocytes were scattered in the Hi and molecular layer (ML). (**K**) In P10 SGZ, Tbr2+ IPs clustered near Sox9+ NSCs with GFAP+ radial processes (arrows). (**L**) GFAP+/Sox9+ astrocytes, including mitotic forms (arrows), were also present (shown in ML).

The online version of this article includes the following figure supplement(s) for figure 1:

**Figure supplement 1.** IPs, NSCs and Notch signaling during DNe formation, developmental migrations, and adult DG expression.

**Figure supplement 2.** The DNe is molecularly and spatially distinct from the early hem/fimbria.

**Figure supplement 3.** Progenitors are distributed throughout transient outer niches by mid-to-late embryonic stages.

Given the relationships between IPs, NSCs, and Notch signaling, we examined how different Notch signaling components were expressed in the early DNe compared to neighboring stem zones. We previously found IPs were a major source of Notch activating ligand Dll1 in the neocortex (*Nelson et al., 2013*). Like *Eomes* expression, *Dll1* expression also was highest in the DNe (*Figure 1A*, *Figure 1—figure supplement 1C*). Overlapping *Notch1* and *Notch3* receptors and *Hes1*, *Hes5*, and *Hey1* effector gene expression occurred in E11.5 DNe (*Figure 1—figure supplement 1C*). Interestingly, high *Hes5* expression also marked the early DNe, likely due to focally increased *Lfng* expression, a key cofactor necessary for processing mature Notch receptors (*Semerci et al., 2017*; *Figure 1—figure supplement 1C*), indicating high overall Notch signaling activity in the DNe. By E13.5 however, downregulation of *Hey1* and concomitant transient upregulation of *Hey2* in the DNe was observed, which may be in part due to developmental shifts from *Notch3* to *Notch2* expression (*Figure 1—figure supplement 1C*), suggesting multiple temporal combinatorial levels underlie high Notch signaling in the early DNe.

We next studied the initial stage of DMS formation and cell migration from the DNe, focusing on IP and NSC interactions, especially Notch signaling (*Hodge et al., 2013*; *Nelson et al., 2013*; *Scott et al., 2010*). Dll1 immunolabeling revealed intense Dll1 protein expression in the DNe at E12.5, validating high *Dll1* transcription levels (*Figure 1B*). We also took advantage of a BAC transgenic GFP mouse line to visualize Tbr2 IPs with their processes to better localize Dll1+ puncta: Tg (*Eomes-EGFP*)DQ10Gsat, simplified to Tbr2GFP (TG) (*Kwon and Hadjantonakis, 2007*; *Nelson et al., 2013*). Increased Dll1+/Tbr2+/Tbr2GFP+ and Dll1+/Dll3+ immunolabeling confirmed early IP production and accumulation in the DNe and overlying SVZ compared to hem and hippocampal CA field stem zones (*Figure 1B*). Dll3 is a Dll1 homolog marking later stage IPs in neocortex that have begun downregulating Dll1 as IPs further differentiate (*Ladi et al., 2005*; *Nelson et al., 2013*). Sox9 showed NSCs in the DNe but not the DMS, confirming that IPs and not NSCs initially accumulated in the E12.5 SVZ (*Figure 1B*). Hes1 protein levels were relatively high in DNe NSCs, and were higher in non-mitotic NSCs compared to mitotic NSCs andTbr2GFP+ IPs (*Figure 1B, B1*), similar to the neocortex (*Mizutani et al., 2007*; *Nelson et al., 2013*). Punctate Dll1 was expressed in Tbr2GFP+ IPs and was notably enriched in their VZ processes, particularly near the ventricular surface (*Figure 1B1*). NSCs (Tbr2GFP-) exhibited various Hes1+ levels (*Figure 1B1*), consistent with heterogeneous, reiterative, and oscillatory signaling (*Imayoshi et al., 2013*; *Nelson et al., 2013*; *Shimojo et al., 2016*). Interestingly, within the intensely Dll1-labeled DNe, Hes1-/Dll1+/Tbr2GFP+ IP mitoses were observed in outer VZ/SVZ regions, and further distinguishing the DNe from adjacent stem zones at this stage (*Figure 1B1*). While transgenic labeling allowed us to determine that IPs and their GFP+ processes are a main source of Dll1, much Dll1 puncta was present in neighboring Tbr2GFP- progenitors in the DNe, consistent with its oscillatory regulation and suggesting additional NSC heterogeneity and diversification prior to differentiating into the IP fate (*Kawaguchi et al., 2008*; *Nelson et al., 2013*).

## IPs pioneer the DMS leading NSCs through multiple outer transient zones

While a previous study indicated numerous transgenically labeled NesGFP+ Pax6+ NSCs present in the early DMS (*Li et al., 2009*), we used more specific molecular markers to identify and locate IPs and NSCs, and by E14.5, the DMS became distinct as a radial extension of the SVZ to the subpial zone (*Figure 1C*). At this early stage, the DMS contained abundant Tbr2+ IPs, but very few Sox9+

NSCs (*Figure 1C,D*). Most DMS cells were IPs that co-expressed Tbr2 and Pax6 (the latter indicating early-stage IP identity), although a few NSCs (Tbr2-/Pax6+) were seen in the proximal DMS (*Figure 1D*, *Figure 1—figure supplement 2*). Hence, early DG IPs transiently express Pax6, similar to neocortical IPs (*Englund et al., 2005*; *Nelson et al., 2013*), and likely retain GFP expression from transgenically labeled NesGFP+ NSCs, obfuscating clear IP versus NSC identification. In the subpial zone, the early DMS ran parallel to, but just beneath the marginal zone, where Cajal-Retzius neurons (CRs, *Trp73+*) migrated from their source in the fimbria towards the hippocampal fissure (*Figure 1—figure supplement 1B* and *2*; *Hodge et al., 2013*; *Yoshida et al., 2006*). By E15.5, *Eomes+* IPs accumulated distally to form the DG, just proximal to the hippocampal fissure, by contrast to the DG anlage, which contained very few NSCs (*Sox2+*) (*Figure 1—figure supplement 1B*). However, by E16.5, Sox9+ NSCs were distributed throughout the DMS and some reached the DG anlage (*Figure 1E*). Together, these data suggested that the DMS is pioneered by an early cohort of IPs, which seemed to 'pave the way' for subsequent NSC migration.

From E16.5 to postnatal day 10 (P10), progenitors continuously, migrated, occupied, and divided in multiple outer niches including the DMS, the fimbriodentate junction (FDJ), the subpial neurogenic zone (SPNZ), the hilus (Hi), and the subgranular zone (SGZ) (*Figure 1A*; *Altman and Bayer, 1990a*; *Angevine, 1965*; *Hodge et al., 2013*; *Li et al., 2009*). By E16.5, multiple outer niches (DMS/FDJ/SPNZ/SGZ) were distinguished by IP GFP reporter mice, in this case the knockin *Eomes*$^{GFP/+}$ line (*Arnold et al., 2009*), simplified to Tbr2GFP (KI) (*Figure 1—figure supplement 3A-B*), including abundant mitotic Tbr2+ IPs and Sox9+ NSCs (*Figure 1E*). The nascent SGZ was marked on E16.5 by the presence of horizontally aligned Tbr2+/Tbr2GFP+ IPs, located just beneath the row of Prox1+ GNs in the newly forming suprapyramidal blade (SPB) of the GCL (*Figure 1—figure supplement 3A-B*). By E18.5, the expanding SPB was surrounded by multiple contiguous outer niches (DMS, FDJ, SPNZ, Hi, SGZ) (*Figure 1F*, *Figure 1—figure supplement 3C*). While NSCs and IPs were located in the overlying SPNZ and appeared to migrate through the nascent SPB, some had accumulated in the nascent SGZ (*Figure 1F,G*). Co-labeling of Sox9+ NSCs with PCNA, a marker of proliferating cells, showed many NSCs in the SPNZ and SGZ were Sox9+/PCNA+, while some were Sox9+/PCNA-, indicating a quiescent NSC state, especially along the SGZ (*Figure 1G*). While many Nestin- and/or GFAP-labeled glial processes (and some mitotic figures) were aligned along the DMS, radial processes uniformly spanning the SPB were not yet apparent (*Figure 1—figure supplement 3C*).

By postnatal day 2 (P2), at the peak of GN neurogenesis, the infrapyramidal blade (IPB) of the GCL had begun to form adjacent to the FDJ, and all outer niches contained abundant IPs and NSCs (*Figure 1H*). Many mitotic figures were seen, for example, Tbr2+ IPs dividing in the nascent SGZ (*Figure 1H*) and Tbr2+/Sox+/Pax6+ mitotic figures (transition from NSC to IP) in the FDJ (*Figure 1I*). By P10, the GCL displayed adult-like morphology, and the SGZ was distinct with abundant Sox9+ NSCs and Tbr2+ IPs (*Figure 1J*). Many Sox9+ NSCs in the SGZ exhibited a prominent single GFAP+ radial process that branched in the GCL (*Figure 1K*) like 'type I' NSCs in adult hippocampus (*Nicola et al., 2015*). Conventional astrocyte (multipolar) proliferation was also active around P10 (*Figure 1L*). From P10 on, IP and NSC proliferation gradually decreased in outer transient niches until P21-P28, when the SGZ became the only remaining outer niche (not shown).

These data suggest that outer neurogenic niches in the developing DG are highly dynamic, and essentially progress by migration of IPs followed by NSCs. The SGZ arises as part of this process, and contains not only IPs but also NSCs (both proliferative and quiescent) at mid-to-late embryonic periods in mice. The primary role of IPs in formation of the DMS and other outer niches further suggested that IPs are required to create a suitable niche for NSC maintenance and proliferation. To investigate this idea, we next tested whether IPs and NSCs interact via Notch signaling in the developing DG.

## Notch-active NSCs follow Dll1-expressing IPs

After initial high Notch activity within the DNe stem zone, expression data indicate progressive shifts in Notch receptors and effectors (*Figure 2A*, *Figure 1—figure supplement 1C*). We examined transgenic *Hes5GFP* reporter mice at E14.5 to visualize NSCs and Notch signaling, and found high reporting in the fimbria and mosaic Hes5GFP labeling in the DNe (*Figure 2B*; *Basak and Taylor, 2007*; *Nelson et al., 2013*). Dll1 immunolabeling of Hes5GFP+ sections revealed punctate Dll1 protein in the DNe VZ and SVZ (*Figure 2B*). Within the DNe, mosaic Hes5GFP marked subsets of mitotic Pax6$_{high}$+ NSCs surrounded by punctate Dll1, and asymmetric divisions were apparent (*Figure 2B1*).

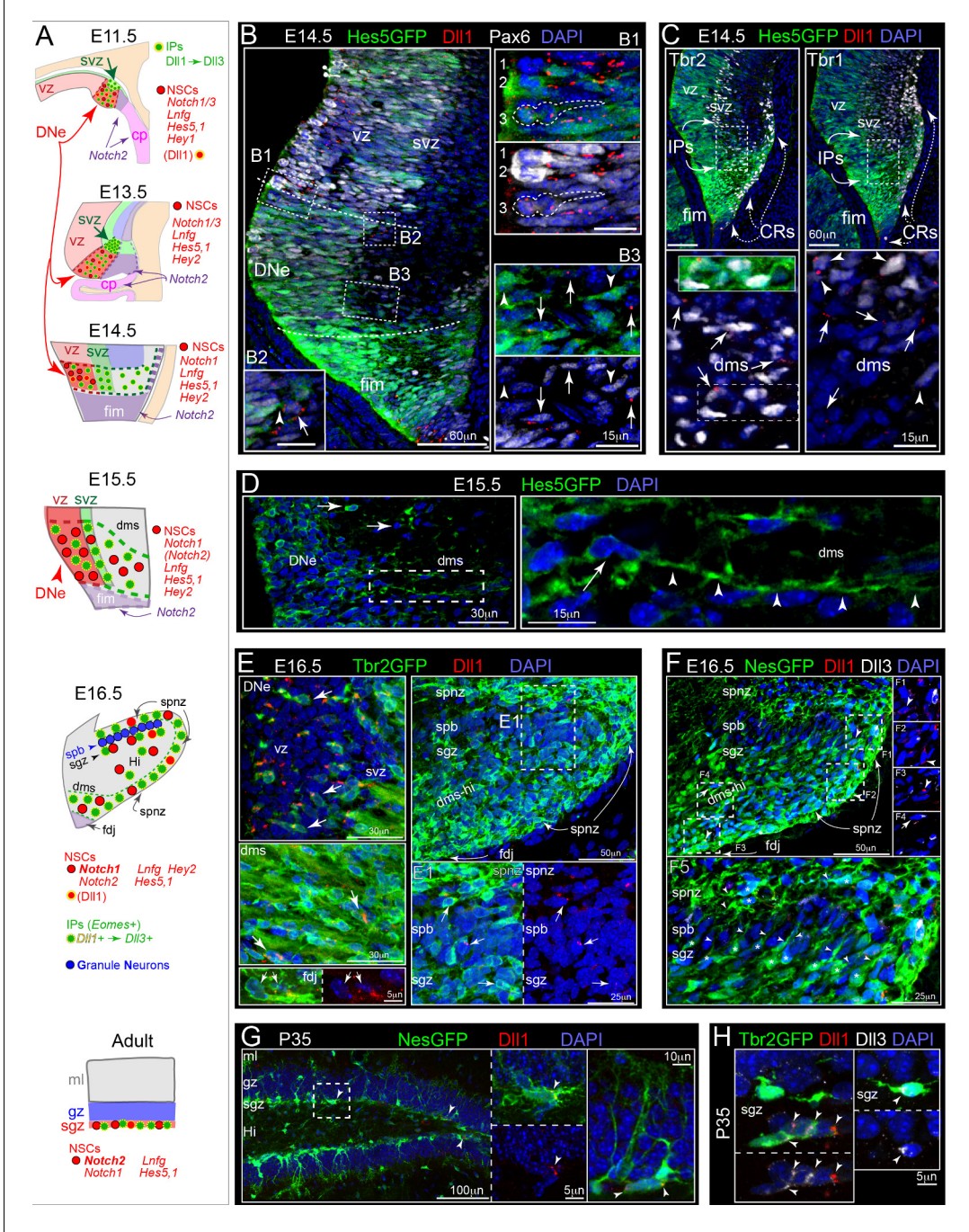

**Figure 2.** Dll1-expressing IPs form a migrating niche for maintaining Notch active NSCs. (**A**) Schematics of DG development and Notch signaling (see also *Figure 1—figure supplement 1*). (**B**) E14.5: Hes5GFP+ coronal sections show high reporting in fimbria, with mosaic Notch activation and Pax6 labeling in DNe (dashed lines): (B1-3) higher power views of boxed regions. (B1) Mosaic Pax6+ and/or Hes5GFP+ (high/low) NSCs dividing at the ventricular surface (NSCs 1–3) near Dll1+ puncta, including asymmetric divisions (NSC 3, Dll1+). (B2) Outer NSC (Hes5GFP+/Pax6+/Dll1-, arrowhead) contacting a Dll1+ IP (Pax6+/Hes5GFP-, arrow) in the SVZ/DMS border. (B3) Sparse mosaic Hes5GFP+ and Pax6+/- NSCs (arrowheads) in the proximal DMS near Dll1+ IPs (Pax6+/HesGFP-, arrows). (**C**) Abundant Dll1+ IPs (Tbr2+, arrows; and some older Tbr1+, arrowheads) in the SVZ-DMS boundary: note some Tbr2+ Dll1+ Hes5GFP+ IPs (inset) can be observed in the initial DMS that runs just below Cajal-Retzius cell (CRs: Tbr2+/Tbr1+, dashed arrows). (**D**) E15.5: Hes5GFP+ NSCs had long processes (arrowheads) and migrated into the DMS. (**E**) E16.5: Dll1+ puncta were present on *Tbr2GFP*+ IPs in the DNe, DMS, FDJ and DG (arrows): E1 higher power view of Dll1+ IPs in the SPNZ and nascent SGZ (arrows). (**F**) E16.5: Some NesGFP+ cells, including NSCs and IPs, exhibited Dll1 and/or Dll3 labeling (F1-4). (F5) NesGFP+ cells, presumably NSCs, with processes aligned radially through the SPB (paired asterisks-arrowheads). Their radial processes (all Dll1-/Dll3-) reached the SPNZ (asterisks-arrowheads). (**G**) P35: NesGFP+ NSCs with radial

*Figure 2 continued on next page*

*Figure 2 continued*

and horizontal morphologies were seen in the SGZ; some horizontal NSCs expressed Dll1 (arrowheads). (**H**) P35: Tbr2GFP+ IPs expressed Dll1 and/or Dll3 (arrowheads).

NSCs (Hes5GFP+/Pax6+/Dll-) in the outer VZ/SVZ contacted Dll1+ early IPs (Hes5GFP-/Pax6+) (*Figure 2B2*). Some Hes5GFP+ NSCs were present in the proximal DMS (*Figure 2B2*), trailing the pioneer Tbr2+ Dll1+ IPs, and distinct from Tbr2+/Tbr1+ CRs generated in the fimbria and overlying migration routes (*Figure 2C*, *Figure 1—figure supplement 2*). Some Tbr2+ Dll1+ Hes5GFP+ pioneering IPs were also observed in the proximal DMS (*Figure 2C*), consistent with previous observations in the neocortex, likely due to rapid cell fate changes and GFP stability (*Basak and Taylor, 2007*; *Nelson et al., 2013*), although IPs due retain proliferative potential that could be further modulated. By E15.5, Hes5GFP+ NSCs were more abundant in the DMS, and some exhibited long leading processes (*Figure 2D*). These data confirm that Hes5GFP+ NSC migrations follow Dll1+ IPs, and illustrate that Notch activation is maintained during NSC migrations to outer regions, a notion supported by scattered expression of *Notch1, Dll1, Hes5*, and *Hey2* in the E15.5 DMS (*Figure 2A–D*, *Figure 1—figure supplement 1*).

By E16.5, many Dll1+ and/or Dll3+ (not shown) Tbr2GFP+ IPs were present throughout all transient outer migration routes and neurogenic niches including the nascent SGZ (*Figure 2E*), although we noticed some Dll1 not associated with IPs. We examined sections from E16.5 NesGFP+ embryos and observed a few Dll1+/NesGFP+ NSCs near the FDJ and SPNZ, while Dll3+/NesGFP+ cells were rare and most likely represent residual GFP protein in IPs (*Figures 2F1–4*; *Hodge et al., 2013*; *Mignone et al., 2004*). Many NesGFP+ NSCs with leading processes were aligned horizontally along the SPNZ (*Figure 2F*). Interestingly, we also observed an array of NSCs located in the nascent E16.5 SGZ with their GFP-filled processes aligned radially through the underlying SPB (*Figure 2*), suggesting some NSCs have already begun transitioning into adult-like morphologies.

We next evaluated Notch signaling in the mature SGZ. On P28, expression of *Dll1, Lfng, Hes5, Notch1*, and *Notch2* were all prominently expressed in the SGZ, indicating a high degree of Notch signaling activity (*Figure 2A, S1C*). On P35, punctate Dll1 immunolabeling was observed on some NesGFP+ cells (NSCs or early IPs) (*Figure 2G*), and on many Tbr2GFP+ IPs (some also Dll3+) (*Figure 2H*). These data indicate: 1) During initial stages of DG development, Notch activity is highest in early DNe compared to other hippocampal and neocortical niches; 2) DNe NSCs have higher Notch activity compared to IPs; 3) IPs are the primary Dll1 source during migrations through outer niches; and 4) adult IPs express Dll1 and thus maintain Notch signaling potential. Altogether, these findings support the idea that high focal Notch activity and production of basally accumulating IPs occur prior to NSC migration to, and expansion in outer regions that lead to morphological changes in DG, as in neocortical gyrus formation (*Llinares-Benadero and Borrell, 2019*).

## IPs divide and migrate through the DMS and outer niches of developing DG in slice culture

We used ex vivo multiphoton live-cell imaging of slice cultures from Tbr2GFP+ reporter mice (TG and KI) (*Arnold et al., 2009*; *Kwon and Hadjantonakis, 2007*) to directly observe the dynamics and migrations of IP cells (*Figure 1—figure*

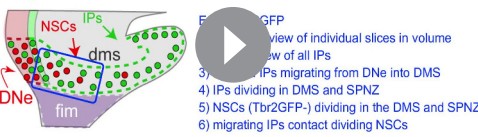

**Video 1.** Emigration from DNe into DMS E16.5 Tbr2GFP+ coronal slices live-imaged during in vitro culture show how typical imaging session is conducted and analyzed, as well as high-power examples of isolated IPs emigrating from the DNe into the proximal DMS, and dynamic IP cell characteristics during migrations, including leading process branching dynaimcs and contacts with neighboring mitotic IPs and NSCs (GFP- visible due to high contrast from IPs) (Tbr2GFP TG). (1) orthoslice view of individual slices in volume (2) volume view of all IPs (3) Isolated IPs migrating from DNe into DMS (4) IPs dividing in DMS and SPNZ (5) NSCs (Tbr2GFP-) dividing in the DMS and SPNZ (6) migrating IPs contact dividing NSCs.
https://elifesciences.org/articles/53777#video1

*supplement 3A, B*; *Source data 1*, *Video 1*; *Hodge et al., 2012*; *Nelson et al., 2013*). Summary maps from each slice were generated to track IPs cell migrations using trajectory vectors in an imaging volume. Also, some isolated cells were digitally surface rendered to reveal IP morphology dynamics during migration (*Figure 3A*; *Nelson et al., 2013*).

In developing DG, as in neocortex, IPs are highly dynamic cells that not only divide and migrate, but also extend and retract filopodia-like leading processes, often contacting other cells. In E16.5 DNe, IPs extended dynamic processes into the VZ (*Figure 3A*, *Video 1*), presumably to interact with NSCs (GFP-), similar to IP→NSC interactions in the neocortical VZ (*Nelson et al., 2013*). Next, IPs in the DNe SVZ withdrew VZ-processes, extended a new outwardly directed dynamically branching leading process, and began migrating into the DMS (*Figure 3A3*, *Video 1*). IPs exhibited coordinated migrations within the DMS, migrating along the fimbria and extending processes into, but never entering this region (*Figure 3A2*, *Video 1*). IPs migrating in the DMS contacted neighboring mitotic progenitors, including other IPs and NSCs (Tbr2GFP-, visible by negative contrast) (*Figure 3A4-6*, *Video 1*). Distally, most early IPs accumulated in the FDJ then continued to migrate and proliferate into the SPNZ, although a few IPs turned from the FDJ into the Hi (*Figure 3A7*, *Video 2*). By E17.5, many IP mitoses were observed in the DMS→FDJ→SPNZ, and in the inner migration route through the Hi, with inner IPs tracking towards outer IPs at the leading edge of the SPB (*Figure 3B1*, *Video 2*). From the SPNZ, horizontally migrating IPs turned inward to migrate radially toward the upper layer of the developing SPB, and IP migration trajectories aligned along the inner edge of the SPB upon reaching the nascent SGZ (*Figure 3B2*, *Video 2*). Interestingly, we observed large-scale morphogenic movements in several slice culture imaging sessions resembling 'gyrification in vitro' (*Figure 4—figure supplement 1*, *Video 2*). By E18.5, IP radial migrations through the SPB, and IP horizontal migrations within the SGZ were even more prominent (*Figure 3C1-2*, *Video 3*). Moreover, temporally distinct developmentally born IP cohorts migrating in different routes (SPNZ→SGZ and DMS→hilus) converged at the SGZ (*Figure 3C3*, *Video 3*). These data reveal the cellular basis for coordinated migration from the DNe into the DMS, and a novel substrate for cell-cell contact dependent signaling mechanisms between migrating IPs and NSCs.

## Live IPs and NSCs interact during migration to the DG

We generated dual color reporter mice lines to directly visualize embryonic IP (Tbr2GFP+) and NSC (NesRFP+) dynamics: *Eomes*$^{GFP/+}$ x *Nes*-CreERT2: *Gt(ROSA)26*$^{Ai14}$ (tamoxifen at E16.0) (*Arnold et al., 2009*; *Imayoshi et al., 2006*; *Madisen et al., 2012*). At E17.5, we observed many GFP+/RFP+ IPs, produced by NSC→IP lineage progression (*Figure 4*), as well as other RFP+ cells (NSCs, vascular pericytes, possibly neurons) likely reflecting complex interactions of TM dose, CreERT2 protein stability, and transgenic *Nestin* promoter expression specificity/level. Nevertheless, 2-color multiphoton live-cell imaging revealed IP (GFP+/RFP+) migrations and IP-IP interactions during mitosis in the DMS→FDJ→SPNZ (*Figure 4A–B*, *Video 4*). Moreover, new views of IP interactions and migrations were seen near developing vasculature (*Figure 4D*, *Video 4*), revealing similar interactions as in developing neocortex (*Stubbs et al., 2009*). Multiple live NSC mitoses (GFP-/RFP+) were observed in the FDJ, SPNZ and the nascent SGZ (*Figure 4A*) with all receiving extensive IP contacts during mitosis (*Figure 4C*, *Video 4*). Orthogonal views revealed many NSCs and IPs had accumulated in the nascent SGZ (*Figure 4E*). Computationally rendering two neighboring IP-NSC pairs in the SGZ showed extensive and prolonged, but likely transient contacts (*Figure 4E,F*, *Video 4*). For example, in one pair (*Figure 4,F1*), as the NSC extended a radial process through the overlying GCL, the IP elaborated its leading process aligned along the SGZ, prior to migrating after prolonged somal interactions (*Video 4*). In the other pair (*Figure 4,F2*), a non-migrating multipolar IP with extensive long-range processes interacted with multiple niche-localized cells, as well as directly with a neighboring mitotic NSC (*Video 4*). We also observed NSCs migrating horizontally in the DMS (*Figure 4G*, *Video 4*), and surprisingly, additional NSCs with leading processes aligned and migrating in a coordinated fashion along the Z-Y axis, in a posterior-to-anterior direction in orthogonal views of coronal slices (*Figure 4H–I*, *Video 5*). To rule out potential slice artifact, we developed a novel *en face* multiphoton live-cell imaging approach (*Figure 4—figure supplement 1*, *Video 5*), mapping anatomical features within the DMS→FDJ→SPNZ region along the longitudinal septal-temporal (anterior/dorsal-posterior/ventral) hippocampal axis. This type of imaging revealed laterally (horizontal) directed IP and NSC migrations through the FDJ→SPNZ route as described,

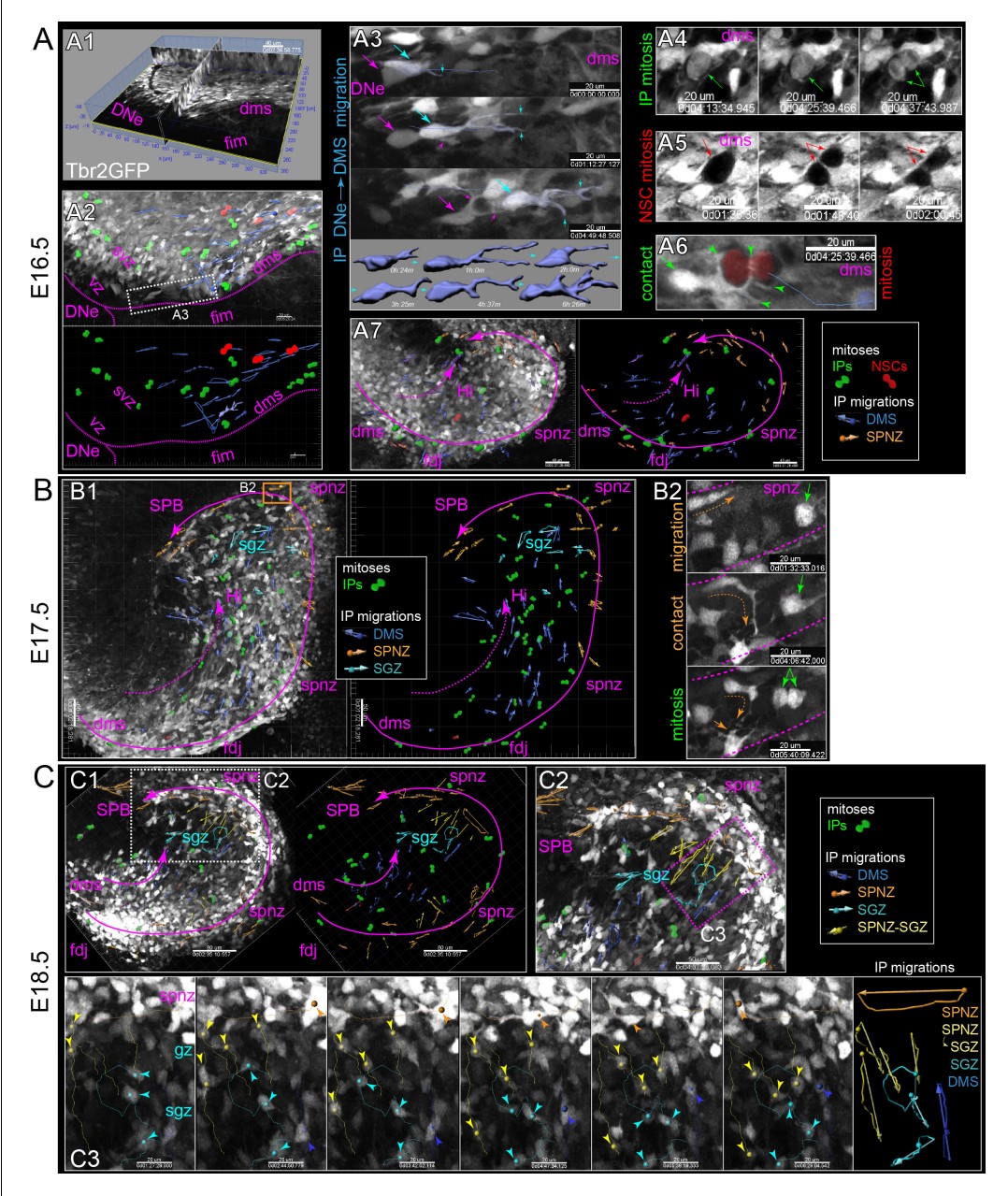

**Figure 3.** Live IPs divide and migrate through developing DG niches. (A) Multiphoton live-cell imaging of Tbr2GFP+ IPs in E16.5 brain slice culture (see also *Video 1*, *Source data 1*). (A1) Grayscale orthoslice view (DNe→DMS). (A2) Volume view of all IPs in DNe→DMS in A1 overlaid with all live IP mitoses (rendered green cells) and trackable migrations (blue vectors): bottom panel summary vector map, some isolated IP cell morphological changes during migration rendered (blue cells). (A3) High power view of sequentially migrating IPs from the DNe SVZ→DMS: note the growth and elaboration of migrating IP leading process (cyan) followed by a second IP with similar migration morphology and dynamics (magenta); bottom shows individual IP cell dynamics. (A4) Live IP mitosis in DMS (green arrows). (A5) Live *GFP-* mitosis (likely NSC) in DMS (red arrows/asterisks). (A6) Live *GFP-* NSC mitosis in DMS encapsulated by multiple IP targeting processes (green arrow/arrowheads). (A7) E16.5 IP mitoses and migrations occurred in outer DMS→FDJ→SPNZ (Tbr2GFP TG). (B) Low power view of Tbr2GFP+ IPs at E17.5 (B1) overlaid with IP mitoses and migration summary vectors: note while IPs migrated in the SPNZ over the SPB, IPs in the DMS→Hi increased and directionally migrated towards SPNZ-derived IPs, and along the nascent SGZ (color coded vectors). (B2) High-power live view of a single IP in the SPNZ migrating backwards, turning inward towards the SPB (orange dashed arrow), and interacting with a multipolar IP (solid orange arrow) near a mitotic IP (green arrows) (Tbr2GFP KI): See also *Video 2*. (C) By E18.5, many Tbr2GFP+ IPs were present in the DMS→FDJ→SPNZ, and IPs migrating directly along the DMS→FDJ→hilus internal route increased (C1-2). (C2) Multiple IP migration routes converge at the nascent SGZ. (C3) Higher power time series of converging IPs at the nascent SGZ (Tbr2GFP KI); see also *Video 3*. The online version of this article includes the following figure supplement(s) for figure 3:

*Figure 3 continued on next page*

*Figure 3 continued*

**Figure supplement 1.** Multiphoton live-cell imaging reveals morphogenesis during early stages of DG formation in vitro.

including many migrations near overlying vasculature (*Figure 4—figure supplement 1*, *Video 5*). *En face* imaging also revealed a sub-population of progenitors aligned longitudinally and migrating near the DMS→FDJ region (*Figure 4—figure supplement 1*, *Video 5*), confirming postero-anterior migration trajectories from coronal slice imaging (*Figure 4H–I*). These 2-color multiphoton live-cell imaging data reveal IPs directly interact with NSCs during migrations through transient outer neurogenic niches to establish the nascent SGZ, and provide evidence for possible additional pathways of NSC and IP migration during DG formation (*Li et al., 2013*).

## Postnatal IPs migrate through outer niches and exhibit extensive process dynamics in maturing DG

By P3, most IPs migrated directly towards the SGZ through the internal DMS→FDJ→Hi route, although proliferative IPs remained in the SPNZ and continued their migrations horizontally around the SPB - IBP and through the GCL to the SGZ (*Figure 5A*, *Video 6*). IP mitoses were more apparent in the Hi-SGZ region, and IPs migrated laterally within the SGZ and vertically into the GCL or Hi (*Figure 5A*, *Video 6*). Thus, converging IP cohorts migrated through outer niches, to continually settle the SGZ. Quantification of different IP groups indicated while IP migration speeds were comparable, IPs in the DMS migrated farther in a directionally coordinated fashion during the imaging duration (*Figure 5B*). Interestingly, IPs in all groups (and ages) exhibited similar 'stop and go' migrations, exemplified by individual IP DMS cell tracks (*Figure 5B*), probing with leading processes while paused, followed by rapid re-engagement of migration machinery. Also, *en face* multiphoton live-cell imaging at P4-5 confirmed a longitudinal orientation shift in IP migrations in the postnatal DMS (*Figure 4—figure supplement 1*, *Video 5*).

In late postnatal DG (P7-P22), IP proliferation and migrations remained evident (*Video 7*), but were increasingly restricted to the SGZ from P12 to P22 (*Figure 5C*). Quantification of all live IP mitoses indicated peak proliferation occurred during late embryonic to early postnatal period (*Figure 5D*). In the more mature P14-P22 DG, IPs also continued to migrate within the SGZ, and extensive transient IP long-range dynamic interactions remained the rule (*Figure 5E1-4*, *Video 7*). IP processes extended distances up to 100–150 µm and displayed complex branching morphologies, presumably interacting with multiple cell types along the SGZ, including mitotic IPs (*Figure 5E1-4*, *Video 7*). Altogether, our live-slice observations open an unprecedented high-resolution window into the cellular basis for DG morphogenesis, and reveal that IPs in the mature SGZ niche continue migratory and dynamic cell-cell interaction behaviors similar to embryonic IPs.

## NSCs require Tbr2 function in IPs to migrate into outer DG neurogenic niches

Since IPs pioneer the DMS and make extensive contacts with NSCs, we sought to determine whether IPs might support NSC migration and suppress NSC differentiation in the DMS. Our previous studies showed that Tbr2 is critical for IP differentiation, function, and survival during DG development and adult neurogenesis (*Hodge et al., 2013*; *Hodge et al., 2012*). Therefore, we analyzed distributions of Sox9+ NSCs within outer niches during early stages of DG formation in mice with genetically ablated IPs (*Nes*-Cre:*Eomes*flox/+), simplified to Tbr2 control and Tbr2 cKO (conditional knockout)

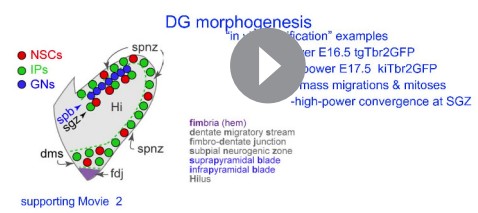

**Video 2.** Morphogenesis in vitro Low-power views of early embryonic. DG slices live-imaged during in vitro culture show morphogenic movements analogous to 'gyrification' in vivo, including large-scale rotations and cell migrations (see also *Figure 3—figure supplement 1*). (1) E16.5 Tbr2GFP TG coronal slice (2) E17.5 Tbr2GFP KI coronal slice.
https://elifesciences.org/articles/53777#video2

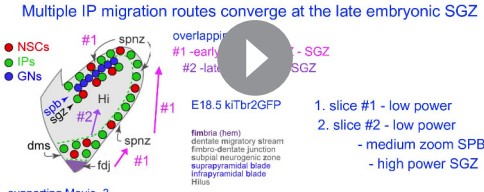

**Video 3.** SGZ convergence E18.5 Tbr2GFP KI coronal slices live-imaged during in vitro culture reveal how multiple different developmentally born IP cohorts migrating though transient niches converge and dynamically settle within the late embryonic SGZ. (1) DMS-FDJ-SPNZ-SGZ (2) DMS-FDJ-Hilus-SGZ. https://elifesciences.org/articles/53777#video3

(*Hodge et al., 2013*; *Intlekofer et al., 2008*; *Tronche et al., 1999*).

At E14.5, while many pioneering Tbr2+ IPs were present in the DMS and accompanied by rare Sox9+ NSCs in controls, only a few Tbr2+ IPs that escaped conditional deletion were present in the Tbr2 cKO DMS (*Figure 6A*). By E16.5, the distribution of Sox9+ NSCs was significantly perturbed in Tbr2 cKO DMS. NSCs appeared disorganized in the proximal DMS, and fewer Sox9+ NSCs reached the distal end of the DMS at the hippocampal fissure (*Figure 6B*). By E18.5, Sox9+ NSCs were abnormally spread across a wide swath of the proximal DMS in Tbr2 cKO DMS mutants, and the distance NSCs migrated from the ventricular surface was markedly reduced compared to controls (*Figure 6C,D*). By P2, the pioneering IP cohort and trailing wave of NSCs had successfully migrated in controls to build the SPB and initial segment of IPB (NeuroD1+) (*Figure 6E*). By contrast, severe SPB hypoplasia was apparent in Tbr2 cKO mutants, NSCs appeared delayed in transient DG niches, and the IPB was conspicuously lacking (*Figure 6E*). These new findings indicate that NSCs enter the DMS but fail to coordinately migrate to the definitive DG, and instead accumulate proximally due to lack of migration cues from Tbr2 cKO depleted IPs in the DMS, followed by their subsequent death and pool depletion (*Hodge et al., 2013*).

## IPs generate GCL neurons in a defined spatiotemporal order

While live-slice imaging revealed how multiple temporally born IP streams arrive at the SGZ and dynamically construct the GCL, this approach could not reveal where the differentiated GN progeny of specific IP cohorts ultimately reside. Hence, we used a lineage specific mouse reporter line, *Eomes*$^{CreERT2/+}$ *x Gt(ROSA)26*$^{Ai14}$ (*Madisen et al., 2012*; *Pimeisl et al., 2013*), simplified to RFP, to precisely determine how sequentially generated embryonic and postnatal IP cohorts contribute to building the GCL (*Mihalas et al., 2016*).

Lineage tracing of early IPs (E13.5) to P0.5 demonstrated extensive labeling of Prox1+ GNs, specifically in the outer shell of the SPB (*Figure 7A, A3*). Significantly, E13.5 IP-derived RFP+ cells did not express Tbr2 or Sox9 (*Figure 7, A1-2*), indicating that none remained as IPs or reverted to NSC identity during the E13.5-P0.5 interval. Similarly, lineage tracing of E16.5 IPs to P0.5 labeled Prox1+ GNs mainly in the outer shell of the IPB, and a smaller number of GNs in the SPB (*Figure 7B, B1-3*). Long-term lineage tracing of early IPs (E13.5-P27) confirmed that pioneering IPs produced GNs in the SPB outer shell, with a sparse distribution reflecting postnatal expansion of the GCL (*Figure 7C*). Lineage tracing from P3 (the peak of DG neurogenesis) to P48 demonstrated abundant RFP-labeled GNs within the middle layers of SPB and IPB, sometimes seen as pairs (*Figure 7D, D1-2*). IP lineage tracing in adult mice (P56 to P84) showed labeled GNs in inner layers of the GCL just outside the SGZ, also sometimes visible as pairs (*Figure 7E*). No glial cells were labeled by IP lineage tracing at any age. Overall, these experiments show that DG IPs exclusively generate all prenatal, postnatal, and adult-born GNs in a defined spatial and temporal developmental order (*Figure 7F*), consistent with previous cell-birthdating and chimera studies (*Angevine, 1965*; *Martin et al., 2002*).

## Discussion

Although the DG is a highly specialized part of the hippocampus, its formation captures many essential and generalizable principles applicable to other migrating progenitor systems in the developing brain, such as neocortical gyrification. These fundamental principles include coordinated migration of IPs and NSCs, maintenance of outer NSCs, and de novo tissue assembly in specific regions. Our results indicate that IPs interact with mitotic and migrating NSCs throughout all stages of DG development, revealing a new cellular substrate for actively transmitting contact-dependent signals, and

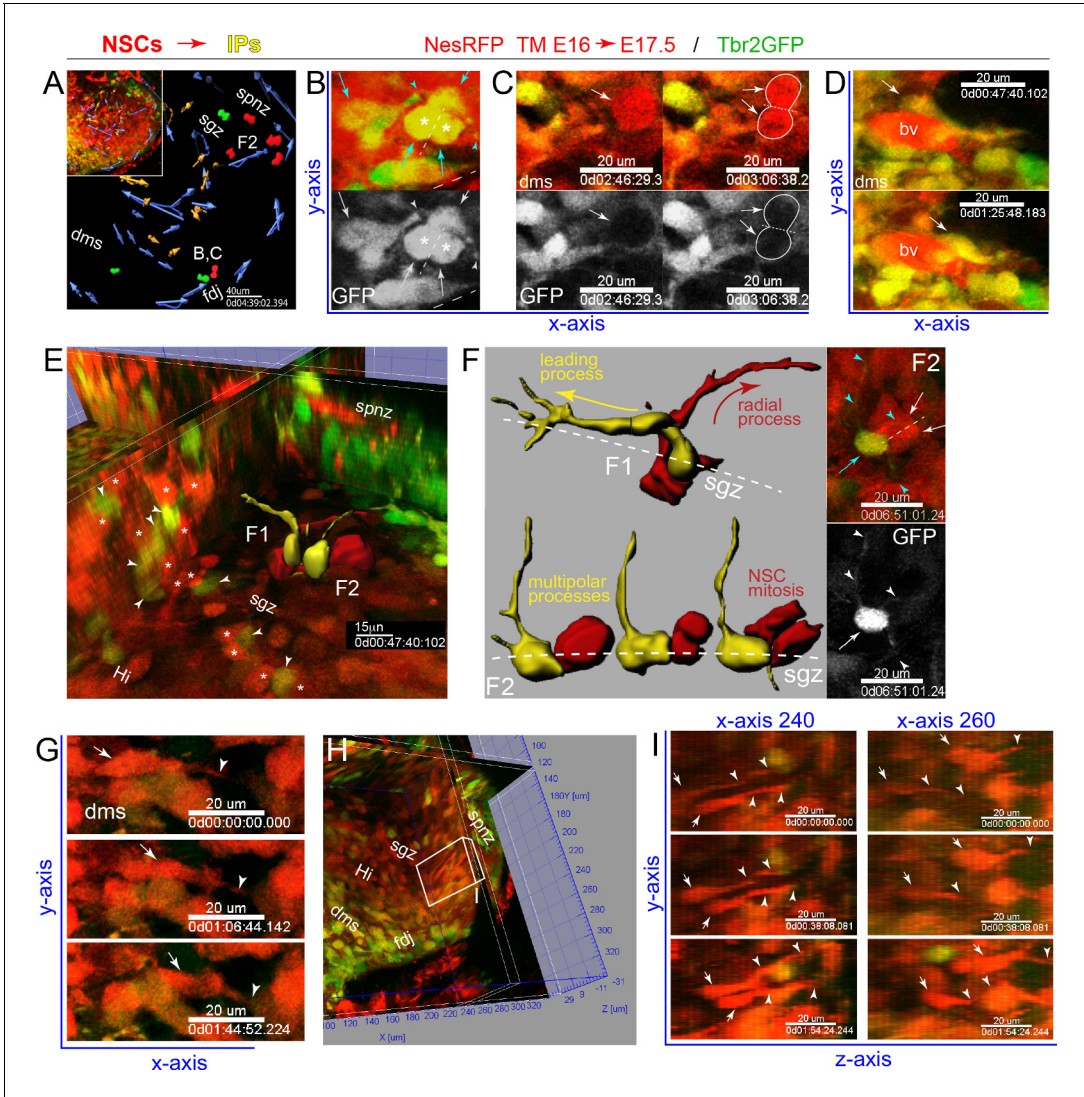

**Figure 4.** IPs and NSCs dynamically interact in developing DG. (**A**) Summary map of 2-color multiphoton live imaging of genetically labeled NSCs (RFP) and IPs (GFP) at E17.5 (A-G, see also *Video 4*). (**B**) High power single optical section views of live migrating GFP+/RFP+ IP with leading process and stationary multi-polar IP (cyan arrows/arrowheads), both interacting with a mitotic IP (asterisks) in the DMS: bottom panel GFP only. (**C**) Two frames showing GFP+/RFP+ IP and leading process dynamically contacting mitotic NSC (RFP+/GFP-, arrows and outline) in the DMS; bottom panel GFP only. (**D**) Two frames showing IPs contacting and migrating near a blood vessel (bv). (**E**) Orthoslice angled to view extensive network of highly juxtaposed IP-NSC cell bodies within the nascent SGZ (IPs, arrowheads; NSCs, asterisks): two IP-NSC cell body pair interactions and dynamics were rendered over time and overlaid in 3D view (F1, F2). (**F**) Rendered IP-NSC cell body pairs isolated and viewed head-on: IPs pseudo-colored yellow (GFP+/RFP+); NSCs pseudo-colored red (GFP-/RFP+). (F1) Cell-cell interactions and IP migration occur in nascent SGZ with NSC extending radial process. (F2) Multi-polar IP with long-range dynamic processes contacting nearby mitotic NSC in nascent SGZ: top right panel, single optical section 2-color live image with IP (cyan arrow/arrowheads) and NSC division (white arrows, dashed line cleavage plane); bottom panel GFP only revealing additional fine processes not captured in rendering (white arrowheads). (**G**) Three frames showing RFP+ NSC with leading process (arrows/arrowheads) horizontally migrating through the DMS→FDJ. (**H**) Orthoslice view along the Y-Z axis within the nascent Hi-SGZ interior of the DG. (**I**) Some RFP+ NSCs with leading processes (arrows and arrowheads) migrated along the Y-Z axis in a coordinated posterior-to-anterior manner (H-I, see also *Figure 4—figure supplement 1* and *Video 5*).

The online version of this article includes the following figure supplement(s) for figure 4:

**Figure supplement 1.** Multiple progenitor migration routes contribute to mosaic DG formation.

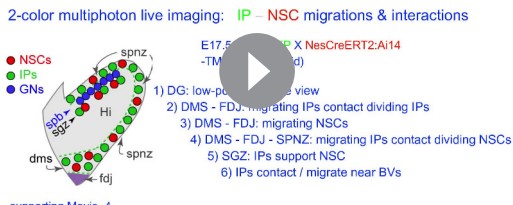

**Video 4.** 2-color multiphoton live imaging: IP – NSC migrations and interactions. Timed matings with TM injection at E16.0 and 2-color multiphoton live-imaging of NSCs (RFP+) and IPs (GFP+) in coronal slices revealed extensive IP-NSC interactions throughout transient niches and in the nascent SGZ: Tbr2GFP KI X NesRFP. (1) DG: low-power volume view (2) DMS - FDJ: migrating IPs contact dividing IPs (3) DMS - FDJ: migrating NSCs (4) DMS - FDJ - SPNZ: migrating IPs contact dividing NSCs (5) SGZ: IPs support NSC (6) IPs contact/migrate near BVs.

https://elifesciences.org/articles/53777#video4

ple Notch signaling molecules (*Notch1/3*, *Hes1/Hey1*, high-level *Lnfg* and *Hes5*) were expressed in the early DNe. Notch-active NSCs (Hes1+/Sox9+) were also abundant in the DNe (*Figure 1*, *Figure 1—figure supplement 1*). Thus, increased genesis of Dll1-production was linked to Notch activa-

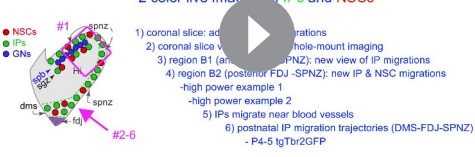

**Video 5.** Additional IP and NSC migration routes contribute to mosaic DG formation −2 color live imaging of IPs and NSCs. Timed matings with TM injection at E16.0 and 2-color multiphoton live-imaging of NSCs (RFP+) and IPs (GFP+) in coronal slices and en face whole-mounts live-imaged reveal additional IPs and NSCs subpopulations longitudinally oriented along anterior/dorsal-posterior migration routes near the DMS-FDJ-hilus/SPNZ contribute to mosaic DG formation: Tbr2GFP KI X NesRFP. (1) coronal slice: additional NSC migrations (2) coronal slice versus en face whole-mount imaging (3) region B1 (anterior FDJ-SPNZ): new view of IP migrations (4) region B2 (posterior FDJ -SPNZ): new IP and NSC migrations -high power example 1 -high power example 2 (5) IPs migrate near blood vessels (6) postnatal IP migration trajectories (DMS-FDJ-SPNZ).

https://elifesciences.org/articles/53777#video5

show that IPs are required to pioneer the DMS and to support NSCs in outer niches. Altogether, these data support a general role for IPs in continuously maintaining NSC pools, prominently through Delta-Notch signaling. More specifically, these data provide a framework for understanding how evolutionarily conserved Notch signaling in migrating NSCs is conveyed at the cellular level through dynamic Dll1-enriched IP filopodia. These findings support the emerging notion that coordinated migration of IPs and NSCs away from the ventricular surface, along with evolutionary modifications in Notch signaling and cell-cell contact, have played important evolutionary roles mediating cortical expansion and morphogenesis.

## The early DNe is marked by increased IP genesis and notch activity

The early DNe (E11.5–13.5; prior to DMS formation) was distinguished by focally high expression levels of *Eomes* and *Dll1* (IP markers), especially in the outer VZ and SVZ. At the same time, multi-

tion in NSCs. Both IP accumulation and Notch signaling are thought to contribute to initial steps in neocortical outward expansion and gyrification (*de Juan Romero et al., 2015*), and we found the same in DG, summarized in *Figure 8A*. Likewise, IPs (especially processes) are the major source of Dll1 in developing neocortex (*Nelson et al., 2013*), a prominent source in the DNe, and the major source during outer DG migrations and formation.

High Notch activity likely reflects the magnitude of rapid cell-fate decisions and proliferation occurring in the DNe to support the burst of Dll1 +/Tbr2+ IP accumulation in the outer VZ and SVZ. These IPs in the early DNe had lower Hes1 levels compared to neighboring DNe NSCs, divided away from the ventricular surface, and extended long-range Dll1+ filopodia back into the VZ as in the neocortex (*Figure 1*; *Nelson et al., 2013*). Interestingly, some Dll1 labeling was not associated with IPs, indicating additional Dll1-Notch signaling operating between diverse NSCs. By E13.5, progressive switches in Notch receptor and effector composition were observed (loss of *Notch3/Hey1*, gain of *Notch2/Hey2*) (*Figure 1—figure supplement 1*) indicating additional temporal layers of combinatorial Notch activities might contribute to regional specification and outer NSC migrations.

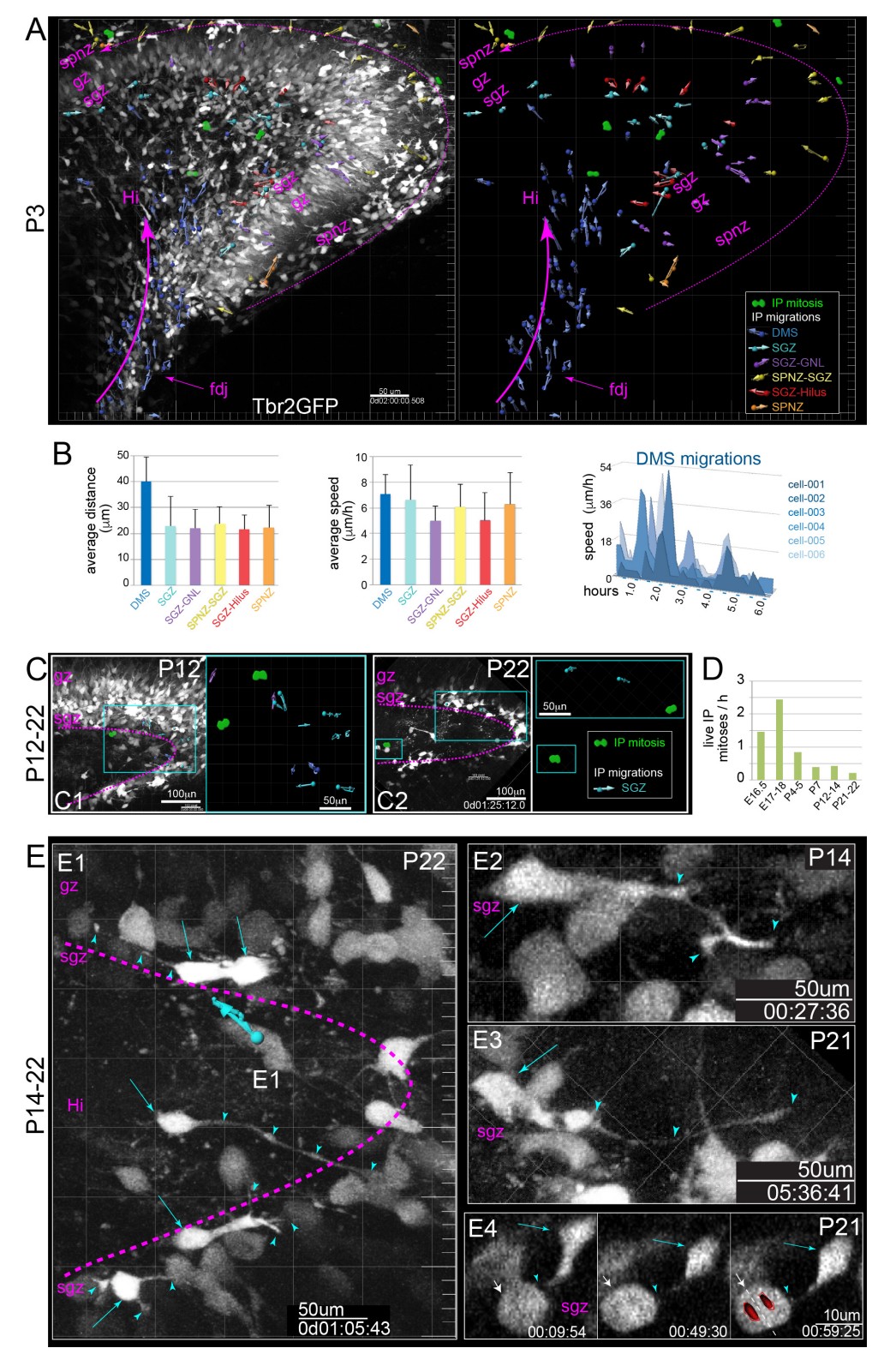

**Figure 5.** IP extensively divide and migrate in postnatal DG. (**A**) Multiphoton live-cell imaging of P3 Tbr2GFP+ brain slices in culture reveal peak period of IP migrations from DMS→FDJ→Hi, extensive IP migrations within the SGZ, and remnants of IP SPNZ→SGZ migrations. IP mitoses were observed within the Hi-SGZ region and remaining SPNZ. Note that maturing GNs in the SPB and IPB remain visible due to high transgenic levels of GFP expression, confirming postnatal Tbr2+ IP→GN lineage (Tbr2GFP TG); see also *Video 6*. (**B**) Quantitation of average IP migration track lengths (left

*Figure 5 continued on next page*

Figure 5 continued

graph) and speed (right graph) through different migration routes at P3: IPs in the late DMS→FDJ→Hi route exhibited sustained and coordinated directionality; individual IPs in the DMS→FDJ→Hi exhibited stop-and-go migration kinetics, consistent with leading process navigation. (C) By P12 (C1) and P22 (C2), IP migrations and mitoses are confined to the SGZ. (D) Quantitation of all live IPs mitoses per hour across time points reveal age-related decline. (E) High power views of IP migrations and dynamics in the late postnatal and young adult SGZ. (E1–E4) Bright GFP+ IPs (cyan arrows) extend dynamic long-range branching filopodia (cyan arrowheads) contacting many cells throughout the SGZ, including neighboring mitotic IPs (E4, white arrow) (see also *Video 7*).

Hence, high combinatorial Notch signaling and increased IP genesis/accumulation in outer VZ regions distinguish the early rodent DNe from other hippocampal germinal zones and may play a role in DNe specification (*Figure 8A*). These results reveal surprising evolutionarily conserved factors and steps marking premigratory zones between DNe formation and neocortical expansion (*Llinares-Benadero and Borrell, 2019*), suggesting further mechanistic insight into mouse DG development, disease, injury and repair can inform human pediatric brain malformations and treatments.

## IPs pioneer NSC migrations into outer developmental zones

Progenitor cells were first demonstrated in the DMS several decades ago (*Altman and Bayer, 1990a*; *Altman and Bayer, 1990b*). More recently, the DMS was shown to contain both NSCs and IPs (*Li et al., 2009*; *Rickmann et al., 1987*), however the pioneer cells of the DMS were not defined, and the scope of interactions between NSCs and IPs was unknown. We sought to determine what type of progenitor(s) first emerged from the DNe stem zone, and found that the DMS was established on E14.5 by IPs migrating first radially away from the ventricle, then tangentially through an almost subpial zone, just beneath the marginal zone (*Figure 1*, *Figure 1—figure supplement 1–2*). The latter contains abundant hem-derived, Reelin-producing CR neurons that also accumulate in the hippocampal fissure, further delimiting DG morphology. These findings suggest that IPs pioneer the DMS to create a suitable environment for NSC migrations into outer niches. By E16.5, NSCs joined IPs in the DMS, and both progenitor types actively divided in multiple transient outer neurogenic niches (*Figure 8B*). The nascent SGZ appeared underneath the Prox1+ SPB as it formed from E16.5–18.5, defined by horizontally aligned Tbr2+/Tbr2GFP+ IPs, radially aligned NesGFP+ NSCs, and the presence of active Sox9+/PCNA+ NSCs and quiescent Sox9+/PCNA- NSCs (*Figures 1–2*, *Figure 1—figure supplement 2*).

Our live-cell imaging directly revealed migrating and proliferating IPs and NSCs transiting through outer niches to reach the nascent SGZ (*Figures 3–5*, *Videos 1–7*). Migration of NSCs was prominent in the embryonic E17.5-E18.5 DG, with live imaging revealing NSCs in the nascent SGZ actively extending radial processes through the GCL, resembling the radial processes of postnatal/adult NSCs (*Figure 4*). Together, these findings raise the possibility that the SGZ is seeded with embryonic NSCs that persist as a source of postnatal and adult neurogenesis. This interpretation is also supported by recent data showing that embryonic DNe NSCs are the origin for at least some adult DG NSCs (*Berg et al., 2019*). These findings suggest that the SGZ is formed even earlier than proposed previously (*Nicola et al., 2015*).

## IPs support NSCs during migrations and new niche formation

The intercellular substrate(s) for transmitting contact-dependent signaling that occurs

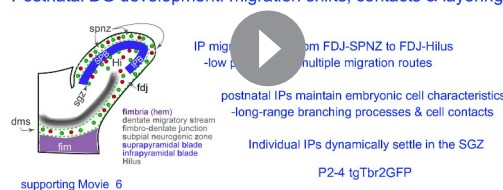

**Video 6.** Postnatal DG development: migration shifts, contacts and layering. Early postnatal Tbr2GFP+ coronal slices live-imaged during in vitro culture reveal predominate shift to the more direct DMS-FDJ-Hilus-SGZ route from earlier DMS-FDJ-SPNZ-SGZ route, and high power views show how IPs dynamically settle within the SGZ (Tbr2GFP TG). (1) IP migrations shift from FDJ-SPNZ to FDJ-Hilus -low power view: multiple migration routes (2) IPs maintain embryonic cell characteristics -long-range branching processes and cell contacts (3) Individual IPs dynamically settle in the SGZ.

https://elifesciences.org/articles/53777#video6

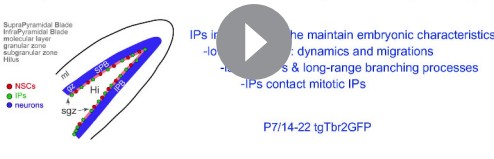

**Video 7.** Young adult DG: maintenance of developmental phenotypes Late postnatal and young adult Tbr2GFP coronal slices live-imaged during in vitro culture reveal cell biological characteristics observed in embryonic and early postnatal DG continue in adult IPs (Tbr2GFP TG). (1) IPs in the late DMS contact mitotic IPs (2) IPs in the adult niche maintain embryonic characteristics -low power view: dynamics and migrations -isolated IPs and long-range branching processes -IPs contact mitotic IPs.

https://elifesciences.org/articles/53777#video7

between diverse and migrating DG progenitors were unclear from previous static models. Dynamic cellular processes can serve as substrates for contact-dependent signals, such as integrin and Notch, and are beginning to be elucidated (*Betizeau et al., 2013*; *Kalebic et al., 2019*; *Nelson et al., 2013*). Although we utilize Notch signaling as our primary molecular focus for context and relationship to neocortical evolution, other cell-contact dependent signaling pathways are also likely being actively transmitted during the dynamic cell-cell interactions we observed in our ex vivo multiphoton live-cell imaging experiments. Importantly, our new live data directly revealed that IPs rapidly target dynamic processes (Dll1+) to neighboring mitotic progenitors (NSCs and IPs) during migrations and in the adult niche. While these intercellular interactions are novel substrates for sending contact-dependent signals, how migrating IPs sense and target their dynamic processes to neighboring mitotic progenitors remains unclear.

By E15.5, Notch component expression appeared in the DMS, revealing Hes5GFP+ NSCs with leading processes in the proximal DMS, and suggesting that NSCs enter the DMS about a day after its inception by IPs. Punctate Dll1 protein was present throughout outer niches during DG development, primarily restricted to Tbr2GFP+ IPs, but also some migrating NesGFP+ NSCs (in SPNZ and SGZ) and NSCs in the SGZ (*Figure 2*). These findings are consistent with reiterative Dll1 usage in sequential neurogenic steps and oscillatory Notch signaling in NSCs (*Kawaguchi et al., 2013*). The interesting switch from *Notch1/3* receptors in early DNe NSCs to *Notch1/2* receptors in migrating and later mature DG NSCs (*Figure 2A*, *Figure 1—figure supplement 1*), was analogous to the evolutionarily similar role of Notch in expansion of human outer NSCs in the neocortex (*Fiddes et al., 2018*; *Hansen et al., 2010*; *Suzuki et al., 2018*). Although conditional genetic ablation of selected Notch components (*Dll1, Notch1, Rbpj, Notch2, Lfng*) demonstrates Notch activity is essential for DG development and/or adult neurogenesis (*Breunig et al., 2007*; *Imayoshi et al., 2010*; *Kawaguchi et al., 2013*; *Semerci et al., 2017*; *Zhang et al., 2019*), phenotypic differences in knockouts along with our data indicate a more complex combinatorial and temporally regulated Notch activity network underlies this process, and that Dll1+ IPs are especially prominent and well positioned to support the bulk of Notch signaling during outer migration of outer NSCs.

Altogether, our findings support the hypothesis that IPs and their dynamic processes serve as a continuous rich Dll1 source, providing robust cell-contact mediated Notch reactivation signals targeting NSCs, maintaining their higher Notch activity and stemness in outer transient niches during DG development (*Figure 8B*), similar to the neocortex (*Kawaguchi et al., 2013*; *Kawaguchi et al., 2008*; *Mizutani et al., 2007*; *Nelson et al., 2013*). Adult DG IPs retain embryonic DG IP characteristics, especially long-range filopodia dynamics and cellular interactions that convey Dll1+ signals throughout the SGZ niche, among other cell-contact dependent signals. Indeed, the loss of IPs in Tbr2 cKO mutant mice, the major Dll1 source during migrations, largely accounts for premature NSC depletion, DG hypoplasia and adult neurogenesis termination phenotypes (*Figures 6* and *8C*; *Hodge et al., 2013*; *Hodge et al., 2012*), similar to conditional Notch inactivation phenotypes. Moreover, DG formation has previously been shown to require the establishment of a glial scaffolding in conjunction with Notch and Reelin signaling (*Brunne et al., 2010*; *Rickmann et al., 1987*; *Sibbe et al., 2009*). We previously found that the glial scaffolding was disrupted in conditional *Nes-Cre/Eomes* based Tbr2 cKO mutants (active at ~E11), whereas delaying deletion several days by using inducible-cKO *Nes-CreERT2/Eomes* mice and tamoxifen injection at E13.5/14.5 largely rescued the glial scaffold, although the DG anlage was still much smaller and premature NSC differentiation and depletion was still observed (*Hodge et al., 2013*). Hence our new data further indicates that

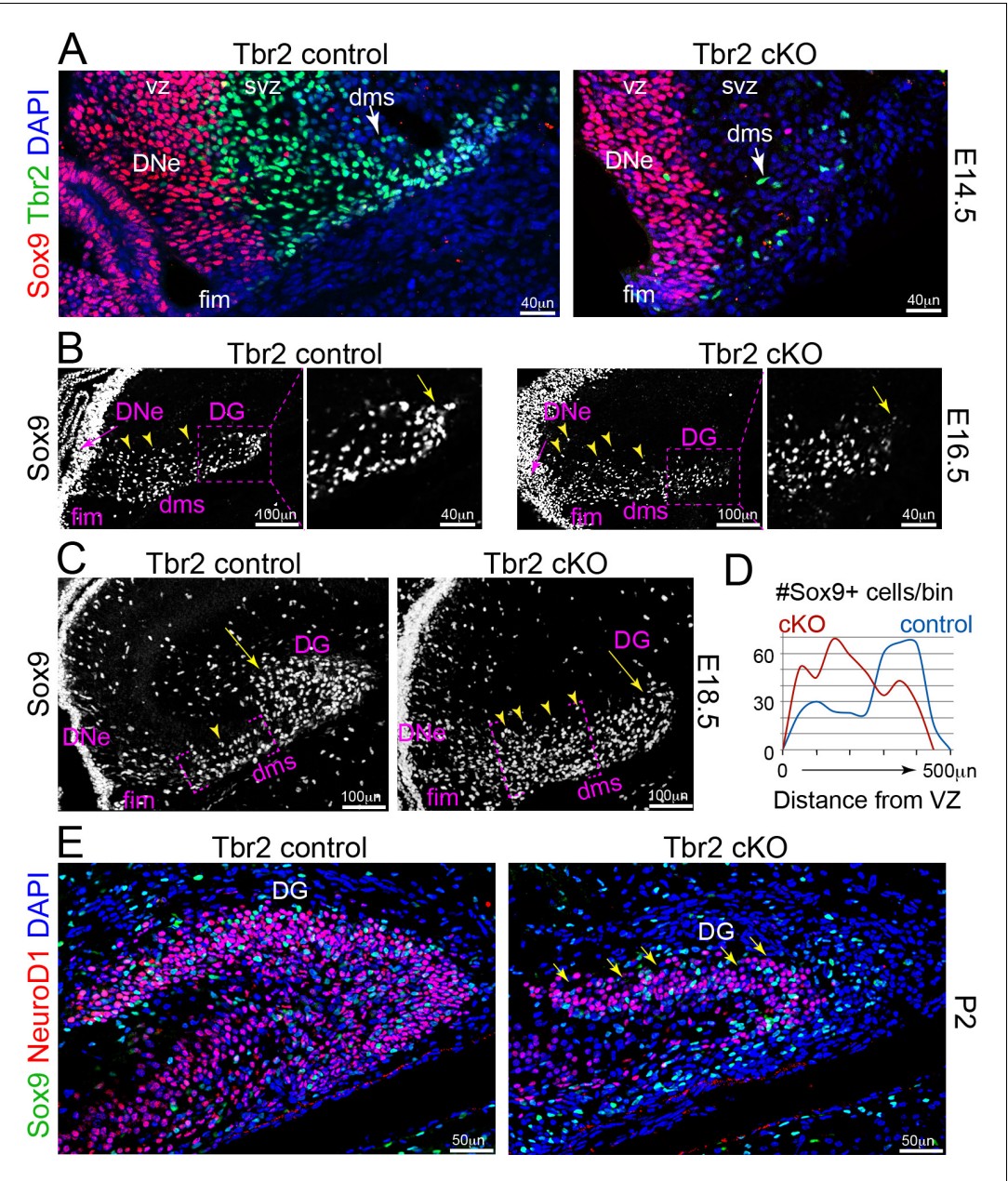

**Figure 6.** IP defects impair NSC migrations and GCL growth in Tbr2 cKO DG. (A) Genetic ablation of IPs: Tbr2 conditional knockout (cKO) versus littermate controls. At E14.5, abundant pioneer Tbr2+ IPs migrated into the initial DMS in controls compared to the few Tbr2+ IPs escaping Tbr2 cKO recombination timing (arrows), while Sox9+ NSCs had yet to emerge from the DNe. (B) By E16.5, Sox9+ NSCs migrated from DNe→DMS, although NSCs in the DNe VZ (arrow), while Sox9+ NSCs in the proximal DMS appeared more dispersed in Tbr2 ckO compared to controls and were less distributed distally (arrow). (C) By E18.5, many Sox9+ NSCs have already migrated to the DG in controls compared to Tbr2 ckO (arrows), which exhibited many more Sox9+ NSCs especially backed up in the proximal DNe→DMS region (arrowheads). (D) Quantification of Sox9+ NSC numbers and migration distance of along the DNe→DMS→DG route reveals impairment in Tbr2 ckO animals. (E) By P2, the Tbr2 ckO DG was severely hypoplastic compared to controls, Sox9+ NSCs and NeuroD1+ new GNs were severely reduced, the SPB GCL remnant was highly dysmorphic (arrows and arrowhead), and the IBP was essentially absent.

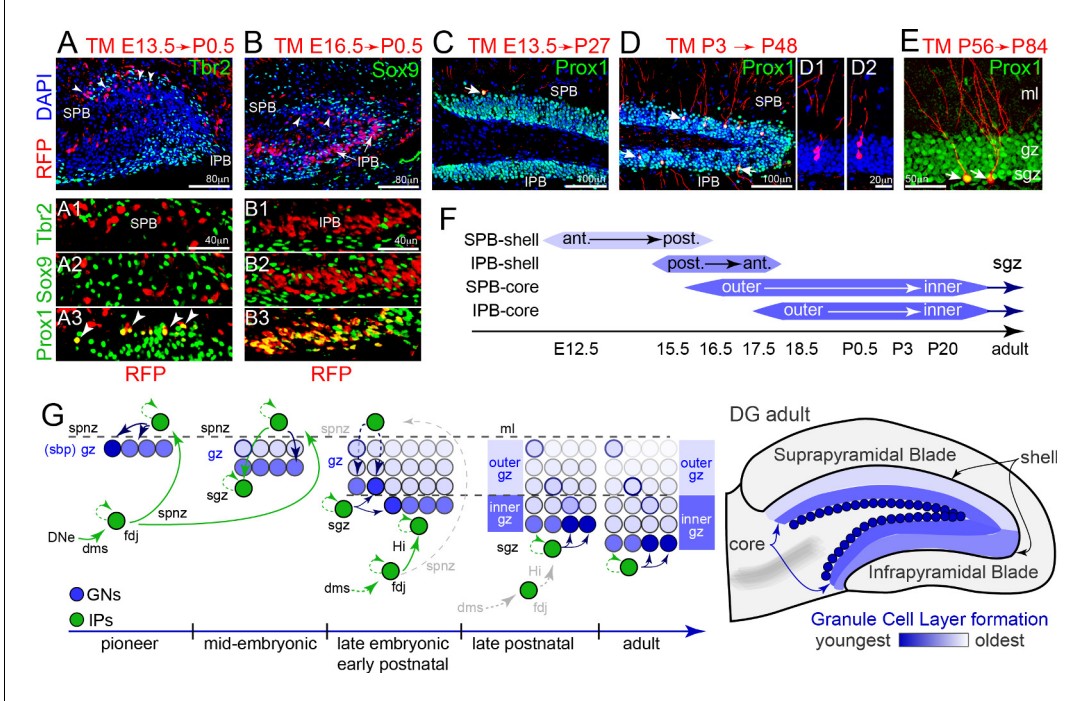

**Figure 7.** Tbr2+ IP cohorts generate GCL shell and core neurons sequentially. (**A**) Pioneer IP cell fates were permanently tracked using single-dose tamoxifen (TM) injection into RFP lineage mouse lines (*Eomes*^CreERT2^x *Gt(ROSA)26*^Ai14^) timed matings at E13.5, followed by immunolabeling for cell type specific markers and position in the DG at P0.5 (sagittal section; left-right, anterior-posterior orientation). Pioneering E13-14 RFP+ IPs were located in the anterior region of outer shell of the P0.5 SPB, and were Tbr2-/Sox9-/Prox1+ GNs (A1-3 arrowheads). (**B**) TM injection at E16.5 labeled some RFP+ GNs in the inner region of the SPB at P0.5 (top panel, arrowheads), and abundant Tbr2-/Sox9-/Prox1+ GNs in the IPB (B1–B3) in a posterior-anterior manner. (**C**) GNs (Prox1+) generated from pioneering E13-14 RFP+ IPs maintained position in the outermost anterior SPB, and exhibited mature GN dendritic arbors and morphologies in the mature P27 DG (arrow). (**D**) GNs (Prox1+) generated from P3 TM-labeled RFP+ IPs occupied middle and outer regions along the P48 adult SPB-IPB, and exhibited mature morphologies (arrowheads): examples of possible sister pairs of GNs (D1–D2). (**E**) Adult RFP+ IPs generated Prox1+ GNs with mature morphologies in the SGZ along inner core edge of the SPB and IPB (arrows). (**F**) Summary of genetic lineage IP→GN cell fate tracing and position in the granule cell layer of the DG. (**G**) Model depicting how pioneering IPs seed GNs in the forming SPB, and how sequentially migrating IP streams converge with new resident IPs at the SGZ to add layers in an outside-inside fashion. (**H**) Generalized granule cell layer birth order and timing from F overlaid onto adult DG schematic.

pioneering Tbr2+ Dll1+ IPs serve as the initial NSC scaffolding, and are also required to help establish the later arising glial scaffolding, in conjunction with Reelin signals from overlying CRs for organizing and maintaining later arising Notch active NSCs, and continue to direct scaffolding changes as the DG matures. How pioneering IPs coordinate NSC scaffolding positions via Reelin and Notch cooperation with CRs remains to be determined.

## Combinatorial notch signaling in migrating versus non-migrating progenitors

Our study, with others, provides a current framework for conceptualizing how combinatorial Notch signaling might function during developmental progenitor migrations and in the adult niche. At the cellular level, in vitro studies (*Khait et al., 2016*; *Shaya et al., 2017*) would predict IP dynamic filopodia with high protrusion forces and increased (punctate) Dll1 loading would generate a strong in vivo Notch activation signal, even with incidental contacts (*Hadjivasiliou et al., 2016*; *Nelson et al., 2013*; *Seo et al., 2016*; *Shaya et al., 2017*). However, our live-cell data indicate IPs undertake more directed interactions targeting contacts to nearby mitotic cells (NSCs and IPs) delivering rapid cell-cell contacts and signals between diverse progenitors. Evolutionary increases in outer neocortical progenitor subtypes (*Betizeau et al., 2013*), as well as increases in progenitor process number (*Kalebic et al., 2019*) are both linked to increased proliferation and expansion into outer brain regions. Hence, the early focal Dll1+ IP accumulation and subsequent pioneering DMS cohort likely

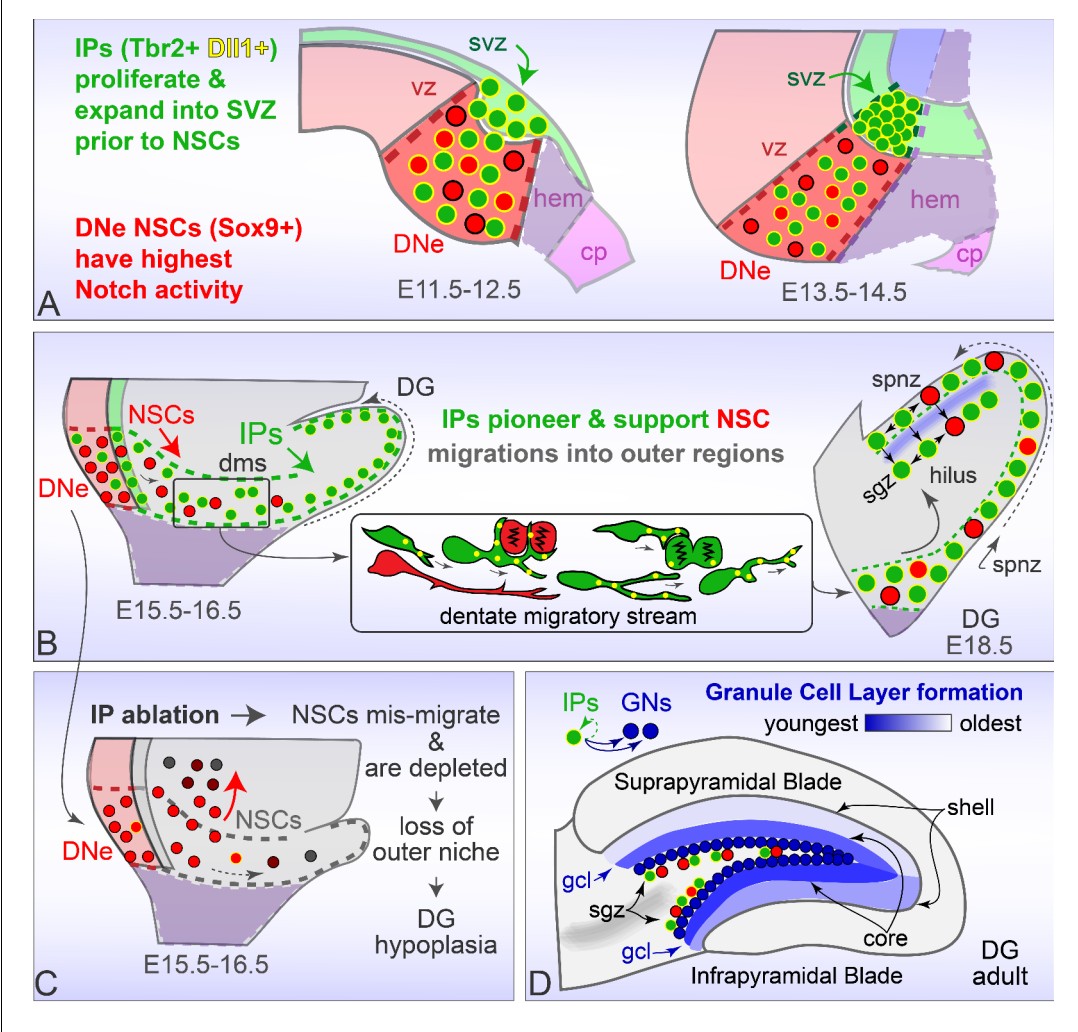

**Figure 8.** Rodent Dentate Gyrus formation. (**A**) Prior to migration, Tbr2+ Dll1+ IPs proliferate and expand into the DNe SVZ, while Sox9+ NSCs are distinguished by high combinatorial Notch activity. (**B**) Dll1+ IPs pioneer migration into outer niches, and target dynamic processes to mitotic neighbors to create a supportive niche for later emigrating NSCs maintenance and guidance. (**C**) Genetic ablation of IPs leads to NSC accumulation and mis-migrations in the proximal DMS, accompanied by rapid NSC depletion in outer regions leading to DG hypoplasia. (**D**). Timed genetic lineage tracing reveals IPs exclusively generate GNs at all ages, and together with multiphoton live imaging reveal how IPs born at different ages transition through different migration routes to converge and layer GNs in a defined spatiotemporal manner.

provide a rich source of dynamic, probing processes mediating contact-dependent proliferation signals to later emerging NSCs, a continuous IP characteristic in the adult SGZ niche (*Figure 8B*).

At the molecular level, Notch receptors in receiving cells can distinguish between different ligands from sending cells (trans-activation), leading to different signaling modes and downstream targets and cell fates (*Nandagopal et al., 2018*). We found at least canonical Dll1 and likely Jag1 (*Nelson et al., 2013*) activating ligands, *Notch1/2/3* receptors, and *Hes1/5-Hey1/2* are expressed during various stages of DG formation (*Figure 2A*, *Figure 1—figure supplement 1*), suggesting different trans-activation scenarios and multiple signaling modes could mediate DNe specification, timing of progenitor migrations, and transient versus permanent niches. Likewise, although our data suggest Dll1+ IPs perform a continuous NSC supportive function, and adult NSCs are derived from embryonic NSCs due to constant lineage specification (*Berg et al., 2019*), migrating and adult NSCs change Notch receptor composition (Notch2, this study) and become activity-dependent (*Crowther and Song, 2014*). These observations indicate although IPs are a continuous Dll1 source,

Notch receptor switching could also change trans-activating signaling modes and outcomes influencing quiescence, proliferation, maintenance, and neuron-astrocyte production during aging.

Recent reports that Notch2 evolutionary modifications underlie outer migrating NSC expansion and diversity in human neocortex (*Fiddes et al., 2018*; *Suzuki et al., 2018*) further underscore how subtle tuning of Notch receptor trans-activation and cis-inhibition (cell autonomous ligand-receptor interaction) can drive activity levels and promote progenitor diversity. Our data show mouse DNe NSCs co-express high levels of Notch receptors and ligands (such as Dll1), predicting potentially strong cis-inhibition, yet we found high Notch activity in DG NSCs. One possible explanation is that oscillatory expression of Notch receptors may itself promote trans-activation between NSCs. In addition, while Mib1 (a key Dll1 co-factor) promotes Dll1 trans-activating Notch signals, Mib1 abrogates Dll1 cis-inhibition to maintain Notch activity in sending cells (*Baek et al., 2018*), suggesting Mib1 could mitigate cis-inhibition in DNe NSCs while generating new IP cell fates. As Mib1 expression appears enriched in IPs, at least in developing neocortex (*Yoon et al., 2008*), Mib1-independent Dll3 expression in late IPs (*Nelson et al., 2013*) could bias towards cis-inhibition and subsequent neuronal differentiation (*Ladi et al., 2005*).

Interestingly, our multiphoton live-cell data show IP dynamic processes also target contacts to neighboring mitotic IPs, even in the mature SGZ niche (*Figures 3–5*, *Videos 1–7*). Hence, while adult DG IPs undergo limited amplification rounds at homeostasis (*Berg et al., 2015*), it is not clear whether developmental IPs are similarly limited. It may be possible to increase IP proliferative potential through targeting increased Mib1/Dll1 IP→IP cell-cell interactions to overcome Dll3 IP cis-inhibition, extending Notch activity and proliferation. Importantly, stimulating one extra IP division would presumably generate a two-fold increase in neurogenic output, and potentially more due to extra Dll1+ IP processes and NSC interactions. On the other hand, IP daughter cells appear to undergo significant asymmetric apoptosis during neocortical development (*Mihalas and Hevner, 2018*) and adult neurogenesis (*Sierra et al., 2010*), which selectively limit neurogenic amplification. While conventional and Cre-dependent conditional/inducible genetic approaches have established the relative importance of many different Notch components (Dll1/3, Jagged1/2, Notch1/2/3, Hes1/5, Hey1/2/L, Rbpj, Mib1, Ascl1, Neurog2), these mutants often suffer from multiple layers of redundancy, lack of progenitor sub-type specificity, and insight into signaling modes (oscillations, amplitude/frequency, force, combinations, locations). As some progenitor types are short-lived (i.e. Tbr2+ IPs), the temporal delay between Cre-dependent expression (i.e. Tbr2) and subsequent recombination events at key Notch loci (i.e. Dll1) may also be difficult to interpret, since in this case, Dll1 would already be expressed in new Tbr2+ IPs, and normally downregulated with Tbr2 as its expression decreases in late-stage postmitotic IPs. Further studies with new and more sophisticated light- and/or drug-inducible perturbations to NSC- and IP-specific cytoskeletal/Dll1-Notch/filopodia dynamics will be necessary to elucidate how these new signaling aspects occur in more in vivo like conditions, contribute to DNe formation, progenitor migrations, and switch to activity-dependent inputs while continuously maintaining DG NSC lineage specification.

## IP migrations generate GCL neurons in a defined spatial and temporal order

Classic birth dating and chimeric studies indicated GNs in the outer GCL layers were born first, followed by GNs in the inner GCL core (*Angevine, 1965*; *Martin et al., 2002*). Our multiphoton live-cell imaging and clonal lineage fate mapping results extend this early work by showing pioneering IPs migrating through the FDJ-SPNZ give rise to the first set of GNs in the SPB (*Figure 7*). As the SPB forms, additional later migrating IPs transit from the overlying SPNZ through the initial GN layer to the nascent SGZ, where their GN progeny contribute to new internal layers. Subsequent IP streams continue to add GN layers, and as NSCs arrive in the nascent SGZ, locally generated IPs also begin contributing GN to internal layers if the SPN. After a temporal delay, the IPB forms similarly with FDJ-SPNZ IPs generating GNs in outer layers first. During later embryonic and postnatal development, IP migrations begin taking a more direct route through the FDJ-Hilus to reach the SGZ, and multiple overlapping developmentally distinct IP cohorts converge at the SGZ to continue layering GNs internally, ultimately leaving the SGZ as the final remaining outer niche. Importantly, our fate-mapping studies revealed embryonic, postnatal and adult IPs are all committed to a GN fate (*Figure 8D*). However, multiphoton live imaging indicates ultimate positioning likely depends on local influences and stochastic interactions, since individual IPs were often observed migrating in

reverse directions and adjusting routes, sometimes even migrating in circles within the nascent SGZ in a dynamic settling process. Hence, local IP positional plasticity during development may serve as a systems-level homeostatic cellular substrate contributing to GCL morphogenic robustness (*Nijhout et al., 2019*). Moreover, we also documented longitudinal migrations (posterior/ventral to anterior/dorsal), a potential cellular basis for ventrally derived embryonic progenitors to reach dorsal DG in the postnatal period (*Li et al., 2013*), and for NSC clonal dispersions (*Berg et al., 2019*; *Li et al., 2013*; *Bonaguidi et al., 2011*; *Pilz et al., 2018*). Thus, multiple IP and NSC sources may mosaically contribute to DG formation.

Given the significance of adult hippocampal neurogenesis, it will be interesting to consider how IP-mediated dynamics and migrations might contribute to memory formation, as well as neurological disorders and injury. Indeed, Tbr2+ IPs are increasingly recognized as important regulators of human brain development, as congenital *EOMES* enhancer mutations cause microcephaly (*Baala et al., 2007*). Moreover, in addition to NSCs, IPs appear to be highly susceptible to environmental teratogen exposure in utero, such as Zika virus, which target IPs in the developing brain and cause life-long consequences for surviving infants including DG abnormalities (*Ho et al., 2017*; *Lin et al., 2017*; *Nelson et al., 2020*; *Adams Waldorf et al., 2018a*; *Adams Waldorf et al., 2018b*). Given emerging roles for IPs in human brain development and health, further studies into IP biology are warranted.

# Materials and methods

## Key resources table

| Reagent type (species) or resource | Designation | Source or reference | Identifiers | Additional information |
|---|---|---|---|---|
| Gene (*Mus musculus*) | *Eomes* | | | *Eomes*→ Tbr2 |
| Antibody | anti-Tbr2 (Rat monoclonal) | Thermo Fisher Scientific | Cat# 14-4875-82, RRID:AB_11042577 | IF (1:250) |
| Antibody | anti-Tbr2 (mouse monoclonal) | Thermo Fisher Scientific | Cat# 14-4877-80, RRID:AB_2572881 | IF (1:500) |
| Antibody | anti-Prox1 (rabbit polyclonal) | EMD Millipore | Cat# ABN278, RRID:AB_2811075 | IF (1:500) |
| Antibody | anti-Sox9 (rabbit ployclonal) | EMD Millipore | Cat# AB5535 | IF (1:1000) |
| Antibody | anti-PCNA (mouse monoclonal) | EMD Millipore | Cat# MAB4076, RRID:AB_95029) | IF (1:500) |
| Antibody | anti-Hes1 (rabbit polyclonal) | Gift from Dr. Nadean Brown (*Nelson et al., 2013*) | | IF (1:1000) |
| Antibody | anti-Pax6 (rabbit ployclonal) | Biolgend (Covance) | Cat# 901301 | IF (1:1000) |
| Antibody | anti-Pax6 (mouse monoclonal) | DSHB | supernatent | IF (1:50) |
| Antibody | anti-NeuroD1 (goat polyclonal) | Santa Cruz | Cat# (N-19): sc-1084 | IF (1:400) |
| Antibody | ranti-GFAP (rabbit polyclonal) | Dako | Cat# GA524, RRID:AB_2811722 | IF (1:1000) |

*Continued on next page*

Continued

| Reagent type (species) or resource | Designation | Source or reference | Identifiers | Additional information |
|---|---|---|---|---|
| Antibody | anti-Nestin (mouse monoclonal) | DHSB | RC2 | IF (1:50) |
| Antibody | anti-Histone H3 (phospho S28) (rat monoclonal) | Abcam | Cat# ab10543 | IF (1:500) |
| Antibody | anti-Reelin (mouse monoclonal) | EMD Millipore | Cat# MAB5364 | IF (1:1000) |
| Antibody | anti-GFP (chicken polyclonal) | Abcam | Cat# ab13970, RRID:AB_300798 | IF (1:500) |
| Antibody | anti-Dll1 (sheep polyclonal) | R and D | Cat# AF5026 | IF (1:100) |
| Antibody | anti-Dll3 (rabbit polyclonal) | Santa Cruz | Cat# sc-67269 | IF (1:100) |
| Antibody | Donkey anti-rabbit ALEXA 488/568/647 | Thermo Fisher Scientific | Cat# R37118, A10042, A-31573 | IF (1:500) |
| Antibody | Donkey anti-rat ALEXA 488/594 | Thermo Fisher Scientific | Cat# A-21208, A-21209 | IF (1:500) |
| Antibody | Donkey ant-mouse ALEXA 488/568/647 | Thermo Fisher Scientific | Cat# R37115, A10037, A-31571 | IF (1:500) |
| Antibody | Donkey anti-chicken ALEXA 488 | Jackson | Cat# 703-545-155 | IF (1:500) |
| Antibody | Donkey anti-sheep ALEXA 568 | Thermo Fisher Scientific | Cat# A-21099 | IF (1:500) |
| Chemical compound, drug | DAPI 4', 6-diamidino-2- phenylindole | Invitrogen | Cat# D1306 | IF (1:10,000) |
| Other | Fluoromout-G | Southern Biotech | Cat# 0100–01 | |
| Chemical compound, drug | tamoxifen (TM) | Sigma | T5648 | |
| Other | NeuroBasal media | Invitrogen | Cat# 21103049 | |
| Other | N2 supplement | Invitrogen | Cat# 17502048 | |
| Other | B27 supplement | Invitrogen | Cat# 17504044 | |
| Other | Pen/Strep | Invitrogen | Cat# 15140122 | |
| Other | FBS | Invitrogen | Cat# 16000044 | |
| Other | HBSS | Invitrogen | Cat # 14025076 | |
| Genetic reagent *Mus musculus* | *Eomes*[GFP] | (*Arnold et al., 2009*) | Tbr2GFP KI | Knockin Tbr2GFP reporter |
| Genetic reagent *Mus musculus* | Tg(Eomes-EGFP)DQ10Gsat | (*Kwon and Hadjantonakis, 2007*) | Tbr2GFP TG | BAC transgenic Tbr2GFP reporter |
| Genetic reagent *Mus musculus* | Eomes[flox/flox] | (*Intlekofer et al., 2008*) | Tbr2 cKO | Cross to Cre lines for inducible and/or conditional knockout |

*Continued*

| Reagent type (species) or resource | Designation | Source or reference | Identifiers | Additional information |
|---|---|---|---|---|
| Genetic reagent *Mus musculus* | *Eomes*<sup>CreERT2</sup> | (*Pimeisl et al., 2013*) | | Inducible CreT2 for tracking IP lineage |
| Genetic reagent *Mus musculus* | *Gt(ROSA)26*<sup>Ai14</sup> | (*Madisen et al., 2012*) | *Ai14* | lox-stop-lox tdTomato reporter |
| Genetic reagent *Mus musculus* | *Nes*-GFP | (*Mignone et al., 2004*) | NesGFP | Transgenic NesGFP reporter |
| Genetic reagent *Mus musculus* | *Nes*-Cre | (*Tronche et al., 1999*) | | Transgenic NesCre driver |
| Genetic reagent *Mus musculus* | *Nes*-CreERT2 | (*Imayoshi et al., 2006*) | | Transgenic NesCre driver |
| Genetic reagent *Mus musculus* | *Hes5*-GFP | (*Basak and Taylor, 2007*) | Hes5GFP | Transgenic Hes5GFP reporter |
| Other | FIJI | NIH | | |
| Other | Photoshop, Illustrator | Adobe | | |
| Other | Imaris 7.4 | Bitplane | | |
| Other | Filmora9 | Wondershare | | |
| Other | Allen Developing Mouse Brain Atlas (2008) | http://developingmouse.brain-map.org/ | | |

## Contact for reagent and resource sharing

Further information and requests for resources and reagents should be directed to and will be fulfilled by the Lead Contact, Branden R Nelson (branden.nelson@seattlechildrens.org).

## Experimental model and subject details

All animals were treated in accordance with IUCAC approved protocols at Seattle Children's Research Institute, kept on a 12 hr light/dark cycle with food and water ad libitum. All of the mouse alleles utilized in this study have been described previously as follows: *Eomes*<sup>flox/+</sup> (*Intlekofer et al., 2008*); NesCre (*Nes11Cre*) (*Tronche et al., 1999*) stock 003771, The Jackson Laboratory; NesCreERT2 (*Imayoshi et al., 2006*), NesGFP (*Mignone et al., 2004*); *Eomes*<sup>CreERT2</sup> (*Pimeisl et al., 2013*); *Eomes*<sup>GFP/+</sup> (*Arnold et al., 2009*); Tg(Eomes-EGFP)DQ10Gsat (*Kwon and Hadjantonakis, 2007*); transgenic *Hes5GFP* reporter (*Basak and Taylor, 2007*); and *Gt(ROSA)26*<sup>Ai14</sup> reporter (*Madisen et al., 2012*): names and crosses are simplified for ease of use as described in the text. All mice were maintained on a C57Bl/6 background, except Tbr2GFP TG (CD1). For timed embryos, healthy fertile adult mice (6–8 weeks old) were paired, vaginal plugs were considered post-conception day 0.5, all embryos were collected at different embryological stages, some were screened for fluorescence, and genotyped accordingly. Postnatal animals were timed, collected and similarly processed. Analyses included mixed sexes of embryos, postnatal, and young adult animals.

## Method details
### Immunolabeling

Immunolabeling was performed on embryonic brain tissue (fixed in 4% PFA 1–4 hr, cryoprotected in 30% sucrose-PBS, cryosectioned at 12 μm, mounted onto slides) (*Nelson et al., 2013*), adult brain tissue (intracardiac 4% PFA perfusion, post-fix 1–2 hr, cryoprotected in 30% sucrose-PBS) was cryosectioned at 40 μm and stained free-floating (*Hodge et al., 2012*), except that adult Dll1/Dll3 immunolabeling was performed on floating vibratome sections (~100 μm). Tissue/slides were washed in

PBS and PBT (PBS-0.01% Triton-X100) 5 min, each step, and then blocked with 10% horse serum in PBT for 1 hr at room temp, with primary antibodies diluted in block and incubated overnight at room temp. Tissue/slides were washed 3-5X with PBT, incubated in species-specific ALEXA 488/568/ 647 conjugated secondary antibody combinations (1:500) in blocking solution (Invitrogen), washed in PBS, counterstained with DAPI (4',6-diamidino-2- phenylindole, Invitrogen), and mounted in Flouro-mount-G (Southern Biotech). Primary antibodies: rat anti-Tbr2 (eBioscience, 1:250), rabbit anti-Prox1 (Abcam, 1:500), rabbit anti-Sox9 (EMD Millipore, 1:1000), rabbit anti-Pax6 (Covance, 1:1000), mouse anti-Pax6 (DHSB monoclonal, 1:50), goat anti-NeuroD1 (Santa Cruz, 1:400), rabbit anti-GFAP (Dako, 1:1000), anti-Phospho-HistoneH3 (rat, AbCam, 1:500); mouse anti-Reelin (CalBiochem, 1:1000); chicken anti-GFP (Abcam, 1:500); sheep anti-Dll1 (R and D, 1:100); rabbit anti-Dll3 (Santa Cruz, sc-67269, 1:100). All sections were analyzed using a Zeiss 710 Quasar 34-channel laser scanning confocal microscope (LSCM, Carl Zeiss). Most images are 1–3 optical Z-sections (~1–3µn), except for *Figure 1* Dll1/Dll3/Tbr2/Tbr2GFP images that are maximum intensity projections (~12 µn). Panels were assembled using Photoshop and Illustrator (Adobe).

## Live-cell imaging and analysis

Acute organotypic coronal brain slices were prepared from embryos (E16.5–18.5) and postnatal animals (P3-22) using approved euthanasia procedures (*Nelson et al., 2013*). Slices were examined for reporter activity (fluorescence stereomicroscope, Olympus), and those with intact DG were adhered onto black nitrocellulose filters (Millipore), stabilized onto cover slides with image seal spacers and a thin coat of diluted low-melting point agarose (~0.1–0.2%), placed into a heated flow-chamber, and imaged under low flow conditions with bubbled media: Neural Basal supplemented with N2 and B27 (Invitrogen), 3 mM HEPES, 0.1% FBS, 5% $CO_2$/95% $O_2$: NB-A was used for postnatal and adult slices; phenol-Red free). Live-cell multiphoton microscopy (MPM) imaging was performed using IR-corrected 25 × 1.05 NA Ultra-objective (Olympus FV1000 MPM), DeepSea MaiTai laser wavelength tuned to 890 nm excitation, image collection began 50–75 µm below the slice surface through a depth of 50–200 µm along the Z-axis depending on the slice, and imaged in 2–3 µm optical slices with Z-stacks collected every 10–12 min (*Nelson et al., 2013*). For postnatal/adult slices, animals were perfused with chilled oxygenated ACSF, and brain slices stabilized same solution until transferred to imaging chamber (*Hodge et al., 2013*), and imaged as described. For *en face* imaging studies, brains were removed, bisected sagittally, subcortical structures were removed to reveal the ventromedial surface of the fimbria, DMS, FDJ, and SPNZ of the IPB, which was positioned and stabilized underneath the objective using low-melting point agarose. Low-power (25X) image volumes were collected along the dorsal-ventral (septal-temporal) longitudinal axis to map the DMS-FDJ-SPNZ IPB region (E17.5 2-color *kiTbr2GFPxNesCreER:Ai14,* P4-7 *tgTbr2GFP*) and time-lapse imaging was performed at higher power on several regions (50X). We required at least 2 successful time-lapsed imaged slices from a given reporter line, with 11 slices over three different lines for mid-embryonic, 5 slices for early postnatal, and 13 slices for later postnatal/early adult stages as detailed in *Source data 1*: even unsuccessful time lapse series (too much drift) provided static confirmation of live cellular orientations and shapes. 4D data sets were processed using Imaris 7.4 (Bitplane) and FIJI (NIH). In each individual case and timepoint, independent sequential 4D datasets were stitched together, drift corrected, and trimmed to generate the longest consecutive and analyzable 4D dataset possible. Full volumes, individual optical sections, and ortho-slices were used to view and analyze cells in 4D datasets. Identifiable cells were manually tracked using Spot function, all observed mitoses as well as some individual IP and NSCs were reconstructed using Surfaces (automatic rendering, refined manually), rotations and sub-volume studies movies and snapshots of regions of interest within 4D volume datasets were isolated using Imaris 7.4 (Bitplane). Exported time sequences and individual timepoint snapshots were assembled using Fiji, Photoshop, and Illustrator, and movies were compiled using Filmora.

## Lineage tracing

To induce recombination, *kiTbr2CreERT2 X Ai14* timed pregnant dams E16.0 were injected with a single dose (100 mg/kg) of tamoxifen (TM) dissolved in corn oil (*Hodge et al., 2013*), and embryos collected were collected at E17.5, screened for fluorescence (GFP+/RFP+, confirmed by genotyping), and processed for 2-color multiphoton live-cell imaging. For *Tbr2CreERT2:Ai14* fate mapping

experiments, timed matings and postnatal/adult animals were injected with a single dose of TM (5 mg/kg), collected at indicated timepoints, and processed for immunolabeling studies with n ≥ 2 embryos/post-natal animals analyzed per time point (*Mihalas et al., 2016*).

## Quantification and statistical analysis

Sample-size estimation for immunolabeling studies were based on our previous experiences, with multiple sections from >3 animals analyzed in immunolabeling studies to determine consistent protein and/or transgenic reporter labeling. All available online ISH sections in developmental each age were analyzed for expression studies (Allen Developing Mouse Brain ISH Atlas). Counts of total observed mitoses per image volume were combined and normalized for each timepoint over respective image durations. Selected live datasets were further analyzed to determine the total number of tracks (identifiable migrating IPs) per group at P3, and statistical analysis of track length over time across groups of migrating IPs was performed using ANOVA and post-hoc Tukey analysis ($p < 0.05$; n = 6 tracks/group). Total Sox9+ NSC counted along the DMS were binned according to distance from proximal DMS to distal DG comparing control versus Tbr2 mutants.

## Acknowledgements

This work was supported by NIH grants R21 MH087070 (RFH, PI and BRN, Co-I), R01 NS085081 (RFH), R01 NS092339 (RFH), R21 OD023838 (BRN and KJM), R01 NS099027 (KJM), and by the German Research Foundation (DFG) through the Heisenberg-Program (AR 732/3–1), project grant (AR 732/2–1), and Germany's Excellence Strategy (CIBSS – EXC-2189 – Project ID 390939984) to SJA.

## Additional information

### Funding

| Funder | Grant reference number | Author |
| --- | --- | --- |
| National Institutes of Health | R21 MH087070 | Branden R Nelson<br>Robert F Hevner |
| National Institutes of Health | R01 NS085081 | Robert F Hevner |
| National Institutes of Health | R01 NS092339 | Robert F Hevner |
| Deutsche Forschungsgemeinschaft | Heisenberg-Program AR 732/3-1 | Sebastian J Arnold |
| Germany's Excellence Strategy | CIBSS - EXC-2189 - Project ID 390939984 | Sebastian J Arnold |
| National Institutes of Health | R21 OD023838 | Branden R Nelson<br>Kathleen J Millen |
| National Institutes of Health | R01 NS099027 | Kathleen J Millen |

The funders had no role in study design, data collection and interpretation, or the decision to submit the work for publication.

### Author contributions

Branden R Nelson, Conceptualization, Resources, Data curation, Formal analysis, Supervision, Funding acquisition, Validation, Investigation, Visualization, Methodology, Project administration; Rebecca D Hodge, Data curation, Validation, Investigation; Ray AM Daza, Prem Prakash Tripathi, Data curation, Investigation; Sebastian J Arnold, Kathleen J Millen, Resources; Robert F Hevner, Conceptualization, Resources, Formal analysis, Supervision, Funding acquisition, Project administration

### Author ORCIDs

Branden R Nelson (iD) https://orcid.org/0000-0003-2941-0153
Kathleen J Millen (iD) http://orcid.org/0000-0001-9978-675X
Robert F Hevner (iD) https://orcid.org/0000-0002-8441-1047

## Ethics

Animal experimentation: This study was performed in strict accordance with the recommendations in the Guide for the Care and Use of Laboratory Animals of the National Institutes of Health. All of the animals were handled according to approved institutional animal care and use committee (IACUC) protocols (#13535) of the Seattle Children's Research Institute.

## Decision letter and Author response

Decision letter https://doi.org/10.7554/eLife.53777.sa1
Author response https://doi.org/10.7554/eLife.53777.sa2

# Additional files

## Supplementary files

• Source data 1. Live-cell multiphoton 4D datasets of embryonic, postnatal, and adult IPs and NSCs in the Dentate Gyrus.

• Transparent reporting form

## Data availability

All data generated or analyzed during this study are included in the manuscript and supporting files.

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
