## [Decision Letter]

**Acceptance summary:**

This work describes how the early migration of intermediate progenitor cells into the dentate primordium, acts as a scaffold for the migration of neural stem cells and formation of the dentate gyrus. The high expression of Dll1 from the filopodia of intermediate progenitors maintain the self-renewal of the underlying stem cell population by activating high levels of Notch signalling. To link the high levels Notch signalling in intermediate progenitors to the formation of the dentate gyrus is an interesting and novel idea.

**Decision letter after peer review:**

Thank you for submitting your article "Intermediate progenitors support migration of neural stem cells into dentate gyrus outer neurogenic niches" for consideration by *eLife*. Your article has been reviewed by Marianne Bronner as the Senior Editor, Francois Guillemot as the Reviewing Editor, and three reviewers. One of the reviewers has agreed to reveal their identity: Lachlan Harris.

The referees have discussed the reviews with one another and the Reviewing Editor has drafted this decision to help you prepare a revised submission.

Summary:

Nelson et al., have studied the development of the dentate gyrus with immunolabeling, genetic lineage tracing and live imaging, and they have examined in particular the role of the intermediate progenitors (IPs) in the ontogenesis of this neurogenic zone. The work describes how the early migration of IPCs acts as a scaffold for the migration of the neural stem cells and the formation of the dentate gyrus. The high expression of the Notch ligand Dll1 from the filopodia of IPs activates Notch signalling in the underlying stem cell population and maintains their self-renewal capacity throughout the extended postnatal period of dentate gyrus development. Live imaging of the developing dentate gyrus is novel and the mapping of the migration of IPs and delineation of how the dentate gyrus develops is done well. The link between expression of Notch ligand by IPs and the formation of the DG is also interesting and novel. The reviewers were overall positive about the study but raised several issues that must be addressed in a revision.

Essential revisions:

1) The most impactful part of the manuscript is the live imaging study that maps the migration of IPs in the DMS and delineates how the DG is generated. However the number of imaged slices is low for such a study where the imaging data are important to support the conclusions. For example, in Figure 1—figure supplement 1, 2 successfully imaged slices seem not enough to make strong statements given the variability of single cell slice behaviour. Key time points should be repeated.

2) Formation of the DG has previously been shown to require the establishment of a glial scaffolding. It would be interesting to find out whether the migration of IPs helps to seed this scaffolding, or whether this is an independent event. This could be determined by looking for the development of the glial scaffolding in TBR2 mutants vs controls.

3) The proposed role of IPs in "continuously maintaining NSC pools, prominently through Delta-Notch signaling" is interesting but the authors show no direct evidence of it. For example, they could delete Dll1 in Tbr2-expressing cells specifically or they could delete Hes5 or some other component of Notch signaling in the NSCs and examine the effect on NSCs migration and behavior in the DMS. Alternatively, the authors should tone down their conclusions on the importance of Notch signaling and focus more on the live imaging part of the manuscript.

---

## [Author Response]

Summary:Nelson et al., have studied the development of the dentate gyrus with immunolabeling, genetic lineage tracing and live imaging, and they have examined in particular the role of the intermediate progenitors (IPs) in the ontogenesis of this neurogenic zone. The work describes how the early migration of IPCs acts as a scaffold for the migration of the neural stem cells and the formation of the dentate gyrus. The high expression of the Notch ligand Dll1 from the filopodia of IPs activates Notch signalling in the underlying stem cell population and maintains their self-renewal capacity throughout the extended postnatal period of dentate gyrus development. Live imaging of the developing dentate gyrus is novel and the mapping of the migration of IPs and delineation of how the dentate gyrus develops is done well. The link between expression of Notch ligand by IPs and the formation of the DG is also interesting and novel. The reviewers were overall positive about the study but raised several issues that must be addressed in a revision.Essential revisions:1) The most impactful part of the manuscript is the live imaging study that maps the migration of IPs in the DMS and delineates how the DG is generated. However the number of imaged slices is low for such a study where the imaging data are important to support the conclusions. For example, in Figure 1—figure supplement 1, 2 successfully imaged slices seem not enough to make strong statements given the variability of single cell slice behaviour. Key time points should be repeated.

We thank the reviewers and agree that our multiphoton live-imaging is quite impactful for revealing the extreme cellular dynamics and migrations underlying DG formation for the first time. We clarify our statement regarding slice number such that:

“at least 2 successful slices from a given reporter line, with 11 slices over three different lines for midembryonic, 5 slices for early postnatal, and 13 slices for later postnatal/early adult stages as detailed in Table 1: even unsuccessful time lapse series (too much drift) provided static confirmation of live cellular orientations and shapes.” (Materials and methods section).

While the number of slices at single time points is not a lot, the coverage across multiple ages, along with our extensive histology series, supports our conclusions. Also, further time-lapse slice experiments are not practicable because the research team has dispersed (Dr. Hevner is now at UCSD), some of the mouse lines are no longer at Seattle Children's, and our multiphoton system at Seattle Children’s is currently being reconfigured for whole animal imaging: ex vivo preps will then have to be re-optimized on a new stage insert.

2) Formation of the DG has previously been shown to require the establishment of a glial scaffolding. It would be interesting to find out whether the migration of IPs helps to seed this scaffolding, or whether this is an independent event. This could be determined by looking for the development of the glial scaffolding in TBR2 mutants vs controls.

This is an interesting point that we have previously addressed (Hodge et al., 2013), where we used conditional (Nes-Cre E11àonward) versus inducible conditional (Nes-CreERT2 with tamoxifen at E13.5/14.5) to temporally delay Tbr2 deletion. We found that the glial scaffolding was disrupted in NesCre based mutants, whereas delaying Tbr2 deletion largely rescued the glial scaffold, although the DG anlage was still much smaller and premature NSC differentiation was still observed. In light of our new data, this indicates that pioneering IPs are also required to help establish the initial glial scaffolding for later arising NSCs, IPs, and early differentiating neuroblasts. We have noted this point in the Discussion section.

3) The proposed role of IPs in "continuously maintaining NSC pools, prominently through Delta-Notch signaling" is interesting but the authors show no direct evidence of it. For example, they could delete Dll1 in Tbr2-expressing cells specifically or they could delete Hes5 or some other component of Notch signaling in the NSCs and examine the effect on NSCs migration and behavior in the DMS. Alternatively, the authors should tone down their conclusions on the importance of Notch signaling and focus more on the live imaging part of the manuscript.

We agree that testing Dll1 function in IPs is an important next step. While genetic approaches are often viewed as the gold standard, single genetic knockouts in the Notch pathway are notoriously difficult to address due to several issues with redundancy especially confounding, as multiply demonstrated by the Kageyama Lab and others. Using Tbr2-Cre to knockout different components is also an intriguing next step, yet likely will suffer from a major temporal delay due to later Cre-dependent floxing of Dll1, restricting this perturbation to late stage IPs due to their short temporal window of existence, which is when Dll1 is normally downregulated. Future work aims to design more kinetically accurate and regulatable approaches to acutely modulate Dll1 and IP processes directly. Accordingly, we have added a note justifying our focus on Notch for current contextual insight into this increasingly complex signal, and mention of challenges using current generations of genetic tools in the Discussion section.

“The intercellular substrate(s) for transmitting contact-dependent signaling occurs between diverse and migrating DG progenitors were unclear from previous static models. Nevertheless, cell-based contacts are the substrates for contact-dependent signals, such as integrin and Notch, and are beginning to be elucidated (Betizeau et al., 2013; Kalebic et al., 2019; Nelson et al., 2013). Although we utilize Notch signaling as our primary molecular focus for context and relationship to neocortical evolution, other cellcontact dependent signaling pathways are also likely being actively transmitted during the dynamic cellcell interactions we observed in our ex vivo multiphoton live-cell imaging experiments. Importantly, our new live data directly revealed that IPs rapidly target dynamic processes (Dll1+) to neighboring mitotic progenitors (NSCs and IPs) during migrations and in the adult niche. While these intercellular interactions are novel substrates for sending contact-dependent signals, how migrating IPs sense and target their dynamic processes to neighboring mitotic progenitors remains unclear.” (subsection “IPs support NSCs during migrations and new niche formation”).

“While conventional and Cre-dependent conditional/inducible genetic approaches have established the relative importance of many different Notch components (Dll1/3, Jagged1/2, Notch1/2/3, Hes1/5, Hey1/2/L, Rbpj, *Mib1*, Ascl1, Neurog2), these mutants often suffer from multiple layers of redundancy, lack of progenitor sub-type specificity, and insight into signaling modes (oscillations, amplitude/frequency, force, combinations, locations). As some progenitor types are short-lived (i.e. Tbr2+ IPs), the temporal delay between Cre-dependent expression (i.e. Tbr2) and subsequent recombination events at key Notch loci (i.e. Dll1) may also be difficult to interpret, since in this case, Dll1 would already be expressed in new Tbr2+ IPs, and then downregulated as Tbr2 expression decreases in late-stage postmitotic IPs. Further studies with new and more sophisticated light- and/or drug-inducible perturbations to NSC- and IP-specific cytoskeletal/Dll1-Notch/filopodia dynamics will be necessary to elucidate how these new signaling aspects occur in more in vivo like conditions, contribute to DNe formation, progenitor migrations, and switch to activity-dependent inputs while continuously maintaining DG NSC lineage specification.” (subsection “Combinatorial Notch signaling in migrating versus non-migrating progenitors”).